# Associations between youth's daily social media use and well-being are mediated by upward comparisons

Andrea Irmer [1 ✉] & Florian Schmiedek[1,2]

Studies examining the associations between social media use and subjective well-being have revealed inconsistent results and mainly refer to the between-person level. We conducted a 14-day diary study among 200 youths ages 10 to 14 to examine within- and between-person associations of social media use (Instagram, TikTok, and YouTube), subjective well-being (positive/negative self-worth, positive/negative affect), and upward social comparisons (general impression of others being better off). Multilevel structural equation models showed that social media use was linked to lower positive and higher negative self-worth on a daily basis, and that upward social comparisons were linked to diminished subjective well-being on all dimensions. Furthermore, our findings were consistent with (partial) mediation of the effect of social media use on subjective well-being by upward social comparisons on the between- and within-person levels. Youths' feelings that others are better off than them-selves may help explain part of the heterogeneity of previous findings.

[1] DIPF | Leibniz Institute for Research and Information in Education, and Center for Research on Individual Development and Adaptive Education of Children at Risk (IDeA), Frankfurt am Main, Germany. [2] Goethe-University, Frankfurt am Main, Germany. ✉email: a.irmer@dipf.de

Social media have become an indispensable part of people's daily life. Different social media platforms such as Instagram, TikTok, or YouTube are being used at younger and younger ages. Unsurprisingly, parents, researchers, and policymakers are concerned about how the use of social media affects children's and adolescents' well-being. Despite numerous studies on this topic, the evidence remains inconclusive, fueling heated debates on whether social media use is detrimental, inconsequential, or even beneficial for youth's well-being[1–3]. Aiming to draw valid conclusions from the rapidly growing number of empirical studies, several reviews and meta-analyses have been conducted[4–7]. Recently, Valkenburg et al.[8] have published an umbrella review (combining the evidence of meta-analytic studies and reviews) on social media use and adolescents' subjective well-being, highlighting the heterogeneity of studies. They called for within-person studies as well as mediation and moderator analyses that allow to shed light on the potential complex associations of social media use and well-being in youths. The aim of the present study was to respond to this call by examining between- and within-person associations among social media use (i.e., the use of Instagram, TikTok, and YouTube) and four indicators of subjective well-being in children's and young adolescents' everyday lives. Additionally, this study investigated upward social comparisons as a proposed key mediator of the link between social media use and well-being[9,10] and explored potential moderator variables (e.g., self-control failure).

Empirical research on the relation between different types of social media use and different indicators of subjective well-being in adolescents documents a range of positive[11–15], non-significant[16–19], and negative associations[15,20–23]. Most meta-analyses and reviews conclude that, overall, there is a small negative association between social media use and well-being in adolescents[4,5,7,24–28]. However, many researchers have questioned the practical significance of this small average effect and emphasize the heterogeneity of findings[29–31]. A recent umbrella review showed that social media use can be associated with both higher well-being and higher ill-being in adolescents[8]. The authors therefore underline the importance of assessing well-being and ill-being simultaneously, a recommendation that has also been documented in previous research with children[32]. To further explain the heterogeneity of effects, mediators and moderators of the link between social media use and subjective well-being should be identified. As the majority of previous studies refers to cross-sectional data, future studies on social media use and well-being should use methods that allow to distinguish among associations on the between-person and the within-person level[8,33]. Within-person associations are of outstanding importance, because they build the rationale for developing interventions. Specifically, in order for interventions to be effective, the targeted variables (e.g., social media use and well-being) need to be related within individuals. Yet, associations based on differences between individuals (i.e., between-person associations) do not necessarily exist on the within-person level[34]. In fact, it has recently been argued that the associations between social media use and well-being are mainly driven by differences between individuals (i.e., differences in the average levels of variables), while the effects within individuals across time are "small to negligible"[35](p.5). To support or challenge this claim, studies are needed that specifically examine within-person associations among the variables of interest.

Besides, there is a need for research that investigates relations among variables in the natural context of people's everyday lives. However, there are only few studies on social media use and subjective well-being that combine within-person research with assessments in youths' everyday lives. For instance, social media use has been found to be linked to same-day symptoms of inattention/hyperactivity and conduct disorder in 11- to 15-year-olds across a 30-day period of daily assessments[36]. By contrast, there was no evidence for daily technology entertainment (e.g., browsing social media) to be related to mental health symptoms (i.e., symptoms of conduct disorder, inattention/hyperactivity, depression, or worry) in a sample of 12- to 15-year-olds[18]. Likewise, Instagram or social media use were not found to be associated with adolescents' affective well-being[37,38] or self-esteem[39,40] on a within-person level. Aiming to explore the relations in more detail, person-specific effects were examined, that is, individual within-person effect sizes were calculated separately for each adolescent. These analyses revealed differences between adolescents in the significance and direction of the associations between social media use and affective well-being: Most adolescents showed non-significant relations, while some adolescents showed increased or decreased well-being[37,39]. Supporting this heterogeneity, Boer et al.[33] also showed that within-person associations between social media use and life satisfaction ranged from negative to positive across adolescents.

Altogether, previous studies demonstrate that individuals differ in their effects of social media use on subjective well-being. Hence, there is a strong need for research investigating why some adolescents seemingly benefit from using social media, while others are harmed by it and yet others seem to be unaffected[33,38]. As recently emphasized[8], examining mediators and moderators in the associations between social media use and well-being might help to shed light on this heterogeneity. The present study follows up on this by examining upward social comparisons as a mediator of the link between daily social media use and daily subjective well-being in youths and by exploring potential moderators.

Social comparisons constitute a universal human drive and refer to the process of evaluating and learning about the self in relation to other individuals[41]. While they can be beneficial for self-improvement, certain comparison processes can be harmful. As such, upward social comparisons, which refer to the comparison with individuals who are evaluated superior, mostly result in lowered self-esteem and/or worsened mood[42–44]. From around the age of ten, children and young adolescents begin to explore their identity and develop a sense of self[45,46]. Social comparisons usually guide this identity formation process and contribute to either self-confidence and high self-worth or feelings of inferiority and low self-worth[45]. Typically, youths compare themselves primarily with peers of similar age, with a lot of these comparisons taking place in the school context. However, the rise of social media platforms such as Instagram, TikTok, and YouTube has introduced a new form of social comparisons[9]. These platforms allow children and adolescents to compare their lives with those of (mostly older) strangers, so-called influencers, who share curated aspects of their personal lives online. Yet, the world of social media is characterized by a "positivity bias"[47](p.95),[48], referring to the tendency to share mainly positive information online. Hence, content on social media is carefully chosen and tends to portray an idealistic image, with visual content being enhanced using filters that improve facial proportions, for instance[48,49]. Many children and young adolescents are struggling to evaluate whether information on the Internet is valid or not[50]. Consequently, they often perceive other's idealized self-presentations as reality and compare them with their own physical appearance, popularity, or leisure activities[51], which induces upward social comparisons[52]. In line with this, recent research with undergraduates demonstrated that browsing social media primarily triggered upward social comparisons as compared to downward or lateral comparisons that are considered less harmful[51]. Additionally, social comparisons made while using

social media were more extremely upward compared to social comparisons occurring in different contexts, meaning individuals tended to compare themselves to others who were "much better off" than them. These upward social comparisons on social media were found to be negatively associated with state self-esteem and life satisfaction measured after having used social media[51]. Empirical research with adolescents supported this finding, showing that upward social comparisons on social media were associated with diminished well-being (i.e., more depressive symptoms, higher body dissatisfaction, and lower life satisfaction[23,53,54]). Due to the important role of upward social comparisons in the association between social media use and well-being, they have been suggested as a key mediator of respective link in (older) adolescents and adults[9,10,30,55,56]. Between-person research with adults supports this claim. For instance, social media use was shown to be related to feeling worse about oneself via upward social comparisons[51] and passive Instagram use was found to be negatively related to life satisfaction via upward social comparisons[57]. Further studies with (young) adults showed that upward social comparisons mediated the relation between passive social media use (e.g., WeChat and Qzone) and self-esteem[58] and depressive symptoms[59] as well as between Facebook use and self-esteem[60]. However, participants in all these studies were above the age of 17 (mostly university students), leaving unclear how the results would look like in younger samples. To our knowledge, only two studies have examined the potential mediating role of upward social comparisons in the association between social media use and well-being among adolescents. Niu et al.[11] demonstrated that upward social comparisons fully mediated the relation between Qzone use and depression in 12- to 18-year-olds. Yet, in contrast, Boer et al.[23] found no evidence for upward social comparisons to mediate the longitudinal link between problematic social media use and depressive symptoms or life satisfaction in 10- to 16-year-olds[23,33]. Hence, it is still unclear whether upward social comparisons serve as a mechanism linking social media use to well-being in children and young adolescents, particularly on the everyday, within-person level. To address this gap, the present study aimed to examine the associations among social media use, upward social comparisons, and subjective well-being in ten- to 14-year-olds. We specifically focused on this age group for several reasons. First, research indicates that children begin using social media on smartphones around the age of ten[50,61]. Additionally, middle childhood is a critical period for self-development and identity formation[45] and social comparisons become increasingly important[45,46]. Social media platforms provide endless opportunities for comparing oneself to friends or strangers around the world[9]. Thus, our second reason for targeting children and adolescents aged ten to 14 years was to examine the associations between social media use, upward social comparisons, and subjective well-being (e.g., self-esteem) during a developmental period characterized by figuring out one's worth based on comparisons with others. Third, previous research has suggested children to be particularly susceptible to the (negative) effects of media use[62,63], which has also been described as "developmental susceptibility"[64(p.227)]. For instance, high social media use in ten-year-olds has been shown to affect subjective well-being in adolescence, especially in girls[63]. Furthermore, a meta-analysis showed that higher screen time was associated with a higher risk for developing depression in children from ten to 14 years of age, but not for those older than 14 years[62]. Given that children appear to be especially vulnerable to the harmful impact of social media use on well-being and healthy development, intensive research in this age group's everyday lives will enable the strongly needed development of tailored and effective prevention and intervention measures.

Another aim of the present study was to identify person-level variables that moderated the associations between (1) social media use and subjective well-being, (2) social media use and upward social comparisons, or (3) upward social comparisons and subjective well-being. Based on prior work, we examined the following potential moderator variables: sex, self-control failure regarding social media use, and social comparison orientation.

Previous research has pointed to sex differences in the associations of interest in the present study. For instance, female adolescents were found to show a stronger link between technology-based social comparison and feedback-seeking with depressive symptoms than male adolescents[53]. Furthermore, the intensity of social media use at age ten predicted well-being in adolescence for female participants, but not for male participants[63]. Therefore, we included sex as a potential moderator variable, expecting female participants to be more susceptible to the detrimental effects of social media use and upward social comparisons on well-being than male participants.

Besides sex, existing studies with adults also motivated us to examine self-control failure with regard to social media use. Self-control refers to the "ability to override or change one's inner responses, as well as to interrupt undesired behavioral tendencies (such as impulses) and refrain from acting on them"[65(p.274)]. Consequently, self-control failure related to social media use assesses the degree to which individuals use social media platforms although it stands in conflict with other goals or tasks, or with using time efficiently[66]. Social media-related self-control failure is associated with deficient self-regulation[66] and can be predictive of later social media addiction[67]. Previous reviews encompassing individuals of all age groups has indicated that self-control and its failure serve as significant moderators of the effects of media use on well-being[68,69]. Hence, the effect of social media use on well-being depends on individual's degree of self-control failure: Failure to self-control social media use can impair subjective well-being by increasing negative emotions following social media use (e.g., guilt) as well as by decreasing the beneficial impact of social media use, that is, by reducing the experience of positive emotions (e.g., enjoyment, vitality)[68,70]. Therefore, we decided to explore social media-related self-control failure as a potential moderator, assuming children with higher self-control failure to show stronger links between social media use and negative self-worth and negative affect, and to show weaker links between social media use and positive self-worth and positive affect than children with lower self-control failure.

Besides, there is evidence showing that individuals significantly differ in their general tendency to engage in social comparisons and that such differences moderate the effects of social media use on well-being[71,72]. In an experimental study with girls aged 14–18 years, those with a stronger tendency to compare themselves to others were more negatively affected (i.e., reported lower body satisfaction) by viewing (manipulated) Instagram posts than those with a weaker tendency to engage in social comparisons[73]. Similarly, undergraduates with a strong tendency to compare themselves to others were more negatively affected (e.g., reported lower self-esteem) by viewing others' Facebook profiles than undergraduates with a weaker tendency to compare themselves to others[72]. Based on these studies that were mostly conducted with (young) adults, we investigated the general tendency to engage in social comparisons as another potential moderator in our study. Thereby, we expected children with a higher social comparison orientation to show stronger links between social media use and upward social comparisons as well as between upward social comparisons and negative self-worth and negative affect, and weaker links between upward social comparisons and positive self-worth and positive affect than children with a weaker social comparison orientation.

To sum up, the present study examined whether social media use was linked to higher or lower subjective well-being in the everyday lives of youths between the ages of ten and 14 years. Conducting a daily diary study across 14 days allowed us to investigate within-person associations among the variables of interest. We collected data in individuals' daily lives, which provides high ecological validity, because behaviors and emotions are captured in the real world and their associations are investigated under typical conditions in natural contexts[74]. Social media use was operationalized by participants' subjective evaluation of how much ('not at all' to 'very much') they used Instagram, TikTok, and YouTube each day. There is research recommending to separately analyze data referring to different platforms[75]. However, it has also been argued that moving away from single-platform data (i.e., Instagram use only) is essential in order to generalize findings[76]. As the associations examined in this work extend to multi-platform data (i.e., considering the use of different platforms simultaneously), we decided to not analyze single-platform data, but to aggregate data of Instagram, TikTok, and YouTube. These platforms have in common that they focus on image-based information (i.e., photos, videos) and that users can follow and/or be followed in a non-reciprocal way (in contrast to Facebook, for instance). Following recommendations[8,32], we operationalized subjective well-being by positive and negative aspects (i.e., positive and negative affect referring to affective well-being[77]; positive and negative self-worth referring to "a favorable or unfavorable orientation toward the self"[78](p.5)). To further investigate how and why social media use is linked to subjective well-being, upward social comparisons were tested as a mediator. Negative effects of social media use on subjective well-being were expected only if children engaged in upward social comparisons. Unlike prior studies[23,79], we assessed upward social comparisons as the general impression that others have a better life (e.g., are prettier, have cooler stuff) instead of measuring upward social comparisons specifically on social media (i.e., "When I read news feeds (or see others' photos), I often think that others are having a better life than me"[79](p.256)). Thereby, we aimed at extending existing research by examining whether days with higher social media use were days with a more extreme impression of others having a better life, in general. To further explain the heterogeneity documented in previous work, this study aimed at identifying moderator variables that explained why some youths experience stronger (or weaker) associations between (1) social media use and subjective well-being, (2) social media use and upward social comparisons, or (3) upward social comparisons and subjective well-being than others.

## Method

A detailed study protocol, a codebook including the original German items with English translations, and data and analysis code necessary to reproduce the results reported here are available in the Open Science Framework at https://osf.io/cs9um/. This study was not preregistered.

**Participants**. Two-hundred children and young adolescents (103 girls) between the ages of ten and 14 years ($M_{age} = 11.71$, $SD_{age} = 1.02$) and one of their parents (163 mothers) participated in this study. Most of the participating youths ($n = 151$, 75.5%) attended the academic tier of secondary school (Gymnasium) and had German as their native language ($n = 160$, 80.0%). At the time of the assessment, 84.5% (32.5%) of fathers (mothers) were employed full-time, 10.0% (56.0%) were employed part-time, and 4% (10.0%) were unemployed.

**Procedure**. The present data were collected in Germany within the zEbra study between 6 April 2021 and 4 June 2021.

Information on study details were disseminated via social media platforms (e.g., the Instagram account of the authors' institution), e-mails to schools and the federal parents' council, sport and music clubs, and word-of-mouth marketing. The study comprised four parts: a parental questionnaire, a baseline questionnaire, a 14-day diary period, and a post questionnaire. All assessments were implemented as online questionnaires on soscisurvey.de. In the first part, parents were asked to complete a background questionnaire (approximately ten minutes) assessing their child's native language, number of siblings, and their child's personality, for instance. Then, children filled in the baseline questionnaire that started with a video in which we explained the study procedure and instructed participants on how to respond to the items. After having watched the video, children were asked to respond to items on their typical social media use, personality, and well-being, for instance. Completing this part of the study took about 30 minutes. The following day, the 14-day diary period started. During this time, children received a daily e-mail at 7 pm with a link to an online questionnaire. They could access the questionnaire daily from 7 pm to 10 pm and were instructed to complete it (which took about ten minutes) just before bedtime. The day after the diary part of the study, children received a link to a post questionnaire containing similar items as the baseline questionnaire, additional measures such as pathological social media use and a questionnaire on emotion regulation problems, as well as feedback on study participation. This final questionnaire took approximately ten minutes to complete.

Inclusion criteria for study participation were that children owned a smartphone with Internet access and were able to understand the German language. Completing the baseline and the post-questionnaire was compensated with 5€ each. Each completed daily questionnaire was compensated with 1€. If at least 12 (out of 14) daily questionnaires were completed, the amount increased by a bonus of 10€. We obtained written informed consent separately and independently from children and parents. Study participation was voluntary and could be terminated at any time. The study was approved by the Ethics committee of the DIPF | Leibniz Institute for Research and Information in Education (DIPF_EK_2021_11).

**Measures**. The wording and descriptive statistics of all items of daily measures are presented in Table 1.

*Daily social media use*. These items were developed for the zEbra study. Participants reported how much (1 = "not at all" to 5 = "very much") they had used each of the following social media sites on that day: Instagram, YouTube, and TikTok. At the beginning of the study, participants had been instructed to consider only their smartphone use of these platforms. The scale showed good convergence with an objective measure of social media use[80].

*Daily positive and negative self-worth*. Positive self-worth was assessed using four items ("Today, I liked myself just the way I am", "I was completely satisfied with myself today", "I felt really good about myself today", "Today, there was a lot about me that I was proud of") that have previously been used in children[81]. In addition, four items assessing negative self-worth were developed for the present study ("I was disappointed by myself today", "Today, I wish I were different", "I got angry with myself today", "I felt worthless today"). The items were answered from 1 ("not at all true") to 5 ("completely true"). See the Results and Table 1 for a multilevel confirmatory factor analysis and reliabilities on between- and within-person levels.

*Daily affective well-being*. Participants were presented with seven emotional states and were asked to indicate how much they

**Table 1 Descriptive statistics of upward social comparisons, positive and negative self-worth, positive and negative affect, and social media use.**

| Item | M (SD) | M ISD (SD) | ICC | Reliability within/between |
|---|---|---|---|---|
| Upward social comparisons | | | | 0.83/0.96 |
| Today, I had the feeling that others have a better life than me. | 1.56 (1.03) | 0.47 (0.47) | 0.59 | |
| Today, I had the feeling that others are happier than I am. | 1.57 (1.00) | 0.46 (0.45) | 0.61 | |
| Today, I had the feeling that others are more popular than I am. | 1.74 (1.18) | 0.45 (0.46) | 0.70 | |
| Today, I had the feeling that others are prettier than me. | 1.69 (1.13) | 0.44 (0.47) | 0.68 | |
| Today, I had the feeling that others were doing more or cooler things than me. | 1.73 (1.16) | 0.50 (0.49) | 0.65 | |
| Today, I had the feeling that others have more or cooler stuff than me. | 1.65 (1.08) | 0.47 (0.49) | 0.62 | |
| Positive self-worth | | | | 0.86/0.97 |
| Today, I liked myself just the way I am. | 4.23 (1.05) | 0.56 (0.42) | 0.55 | |
| I was completely satisfied with myself today. | 4.09 (1.12) | 0.63 (0.40) | 0.56 | |
| I felt really good about myself today. | 4.01 (1.13) | 0.63 (0.40) | 0.56 | |
| Today, there was a lot about me that I was proud of. | 3.80 (1.27) | 0.77 (0.42) | 0.52 | |
| Negative self-worth | | | | 0.82/0.96 |
| I was disappointed by myself today. | 1.63 (1.00) | 0.64 (0.47) | 0.37 | |
| Today, I wish I were different. | 1.63 (1.04) | 0.58 (0.46) | 0.49 | |
| I got angry with myself today. | 1.63 (1.04) | 0.63 (0.48) | 0.41 | |
| I felt worthless today. | 1.36 (0.84) | 0.40 (0.45) | 0.48 | |
| Positive affect | | | | 0.78/0.94 |
| Today, I felt good. | 4.32 (0.89) | 0.59 (0.39) | 0.39 | |
| Today, I felt fantastic. | 4.07 (1.04) | 0.66 (0.42) | 0.47 | |
| Today, I felt content. | 4.05 (1.11) | 0.77 (0.43 | 0.38 | |
| Negative affect | | | | 0.71/0.94 |
| Today, I felt unhappy. | 1.67 (1.03) | 0.76 (0.48) | 0.26 | |
| Today, I felt sad. | 1.52 (0.92) | 0.65 (0.45) | 0.26 | |
| Today, I felt miserable. | 1.57 (0.95) | 0.68 (0.45) | 0.29 | |
| Today, I felt afraid. | 1.32 (0.76) | 0.43 (0.45) | 0.37 | |
| Social media use | | | | |
| How much did you use Instagram today? | 1.28 (0.72) | 0.18 (0.34) | 0.70 | |
| How much did you use TikTok today? | 1.65 (1.13) | 0.33 (0.45) | 0.75 | |
| How much did you use YouTube today? | 2.17 (1.22) | 0.73 (0.37) | 0.55 | |

$N = 200$. All items were presented in German. Reliability was estimated using McDonald's Omega[102].
ICC intraclass correlation (the proportion of between-person variance to total variance), M ISD mean intraindividual standard deviation.

agreed with each of them that day (1 = "*not at all true*" to 5 = "*completely true*"). There were three items measuring positive affect (good, fantastic, content) and four items measuring negative affect (unhappy, sad, miserable, afraid). The same items assessing momentary emotional states showed good psychometric properties in prior studies with children[32,82,83] and adolescents[84]. In contrast to previous work, the items referred to the whole day instead of a specific moment ("today" instead of "right now") in the present study.

*Daily upward social comparisons.* Based on the work by Boer et al.[23], six items were developed to assess upward social comparisons on a daily basis in youths (e.g., "Today, I had the feeling that others have a better life than me"). The items were answered on a 5-point scale (1 = "*not at all true*" to 5 = "*completely true*"). Hence, higher scores on this scale (i.e., referred to as more extreme upward social comparisons in the following)[51] indicate that participants had the impression that others had a better life than themselves or were happier, prettier, or more popular. That is, they perceived a higher discrepancy between themselves and others. Notably, in contrast to previous studies[23,79], we did not specifically refer to social media in our items but assessed daily upward social comparisons as a general impression of others being better off.

*Sex.* We asked parents to indicate their child's sex and coded it as 0 = male participants, 1 = female participants, and 2 = non-binary participants. However, no parent reported that their child was non-binary, which is why the sex variable was dichotomous in the present study.

*General social comparison orientation.* We used the short German version of the Iowa-Netherlands Comparison Orientation Measure[85] by Gibbons and Buunk[41] and modified the items to make them suitable for assessing social comparison orientation in children. The scale assesses individuals' general tendency to compare themselves with others. Thus, children were presented with three items that formed the "ability" subscale of the comparison orientation measure (e.g., "I always pay a lot of attention to how I do things compared with how others do things") and three items that formed the "opinion" subscale of the comparison orientation measure (e.g., "If I want to learn more about something, I try to find out what others think about it"). The items were rated from 1 ("*not at all true*") to 5 ("*completely true*"). Social comparison orientation was assessed twice, in the baseline assessment and in the post assessment. Due to potential reactivity effects, the values of the baseline assessment were used in the present analyses. The reliability of the scale (i.e., McDonald's Omega) was acceptable (baseline: opinion: .67; ability: .75; post: opinion: .86; ability: .69).

*General self-control failure.* We assessed children's self-control failure in relation to social media use by three items (e.g., "Using social media gets in the way of my other goals, like doing things for school or other tasks"). We translated and slightly adapted items that had been developed by Du et al.[66] and had also been used by Chen et al.[86] in samples of adults. This scale measures how much individuals give in to the desire to use social media even though the use stands in conflict with other goals or demands[66]. We adapted the response scale so that it ranged from

1 ("*not at all true*") to 5 ("*completely true*"). The scale showed an acceptable reliability of .82 in the present study.

**Statistics**. All analyses were performed in Mplus Version 8.8[87]. Our data had a two-level structure, with repeated measures (i.e., days, Level 1) being nested within children (Level 2). First, we conducted multilevel confirmatory factor analyses (MCFAs) to assess psychometric properties of the scales developed and used in our study. We evaluated model fit as acceptable in case the Root Mean Square Error of Approximation (RMSEA) was <.08, the Comparative Fit Index (CFI) was >0.90, and the Standardized Root Mean Square Residual was <0.08.

Then, we performed multilevel structural equation modeling (MSEM) to examine the within- and between-person associations among the variables of interest. In all MSEMs, we used the Bayesian estimator and the Mplus defaults for (uninformative) priors. We employed two Markov Chain Monte Carlo chains with a 50% burn-in, 3000 iterations, and a thinning factor of 50. We report the median of the posterior distribution as a parameter estimate as well as the 95% credible interval of the posterior distribution. Furthermore, we report the model-implied between- and within-person $R^2$ and the 95% credible interval provided by Mplus. Parameters were interpreted as statistically significant in case their 95% credible interval did not contain zero; hence, significance was tested two-sidedly. We modeled all within-person effects as random effects and allowed all random effects and all dependent variables to covary at Level 2. In the first model, we investigated the associations between social media use and positive and negative self-worth as well as positive and negative affect. In the second model, we examined the associations between social media use and upward social comparisons and in the third model, we examined the associations between upward social comparisons and positive and negative self-worth as well as positive and negative affect. For the moderator analyses, a cross-level interaction was included (i.e., the random slope was predicted by the moderator variable) and covariances between the moderator and the predictor variables were allowed.

Furthermore, we estimated a multilevel mediation model in Mplus. Random slopes were estimated for all effects on the within-person level (i.e., the $a_w$-path = the within-person effect of social media use on upward social comparisons; the $b_w$-paths = the within-person effects of upward social comparisons on positive and negative self-worth and on positive and negative affect; and the $c_w$-paths = the direct within-person effects of social media use on positive and negative self-worth and on positive and negative affect). We allowed the four residuals of the dependent variables (positive and negative self-worth, positive and negative affect) and all random effects to covary. The four indirect effects on the within-person level were calculated as $a_w*b_w + cov_{awbw}$, with $cov_{awbw}$ referring to the covariance of respective two random effects. Total effects on the within-person level were calculated as respective indirect effect + $c_w$, with $c_w$ referring to respective within-person effect of social media use on one of the four dependent well-being variables (i.e., positive and negative self-worth, and positive and negative affect). The four indirect effects on the between-person level were calculated as $a_b*b_b$, with $a_b$ referring to the between-person effect of social media use on upward social comparisons and $b_b$ referring to respective between-person effect of upward social comparisons on one of the four dependent well-being variables (i.e., positive and negative self-worth, and positive and negative affect). Total effects on the between-person level were calculated as respective indirect effect + $c_b$, with $c_b$ referring to respective between-person effect of social media use on one of the four dependent well-being variables[88,89].

**Results**

**Descriptive statistics and multilevel confirmatory factor analyses**. Means, standard deviations, intraindividual standard deviations, intraclass correlations, and reliabilities of the main variables can be found in Table 1. The overall compliance rate was good (85%) and slightly higher than in comparable studies[90], yielding 2382 available data points out of a maximum of 2800 data points.

First, we performed MCFAs to examine model fit of the self-worth scales. The items assessing positive self-worth had been used in previous ambulatory assessment studies with children and showed good psychometric properties[81]. The items measuring negative self-worth had been developed for the present study and, thus, had not been evaluated so far. Therefore, we examined whether positive self-worth and negative self-worth could be separated on both, the between- and within-person level. We tested two MCFAs, one MCFA with all eight items loading on one self-worth factor and another MCFA with four items loading on a positive self-worth factor and four items loading on a negative self-worth factor. We then compared the two models using likelihood ratio tests. We used the robust maximum likelihood estimator (MLR), requiring the adjustment of the $\chi^2$-difference test by a scaling correction factor[91]. The analyses revealed that a two-factor model represented the data better than a one-factor model, $\chi^2(2) = 456.82$, $p < 0.001$ (see Supplementary Table 1 for full model results). The correlation of the factors representing positive and negative self-worth was $r = -0.79$, $z = -15.38$, $p < 0.001$, on the between-person level and $r = -0.48$, $z = -7.69$, $p < 0.001$, on the within-person level. Therefore, we decided to enter positive self-worth and negative self-worth as separate factors in all analyses.

We further performed an MCFA to examine model fit and reliability of the scale assessing upward social comparisons that had been developed for the purpose of the present study. Overall, model fit (see Supplementary Table 1) and reliability (see Table 1) were acceptable, indicating that the six items are suited to measure daily upward social comparisons in children and young adolescents.

**Multilevel structural equation models**. All models converged successfully with a maximum probability of scale reduction (PSR) of 1.004. Visual inspection of the trace plots indicated successful mixing.

Figure 1 displays the results of the MSEM investigating the links between social media use and subjective well-being. On the between-person level, social media use significantly predicted all four indicators of subjective well-being. Thus, children and young adolescents who on average used more social media than others across the two weeks of assessment reported lower positive self-worth ($\beta = -0.28$ [$-0.41$, $-0.14$], $R^2 = 7.7\%$ [1.9%, 16.9%]) and positive affect ($\beta = -0.31$ [$-0.45$, $-0.17$], $R^2 = 9.7\%$ [2.7%, 20.1%]) and higher negative self-worth ($\beta = 0.21$ [0.07, 0.35], $R^2 = 4.4\%$ [0.4%, 11.9%]) and negative affect ($\beta = 0.26$ [0.12, 0.40], $R^2 = 6.9\%$ [1.4%, 16.0%]).

On the within-person level, only the effects of social media use on positive self-worth ($\beta = -0.08$ [$-0.13$, $-0.04$], $R^2 = 5.3\%$ [3.3%, 7.7%]) and negative self-worth ($\beta = 0.07$ [0.02, 0.11], $R^2 = 5.4\%$ [3.5%, 7.5%]) were significant, but not the effects of social media use on positive affect ($\beta = -0.03$ [$-0.09$, 0.01], $R^2 = 6.9\%$ [4.7%, 9.5%]) or negative affect ($\beta = 0.04$ [$-0.01$, 0.09], $R^2 = 5.5\%$ [3.7%, 7.6%]). Hence, on days children and young adolescents used more social media than usually, they were less satisfied and more dissatisfied with themselves.

Figure 2 depicts the results of the MSEM examining the associations between social media use and upward social

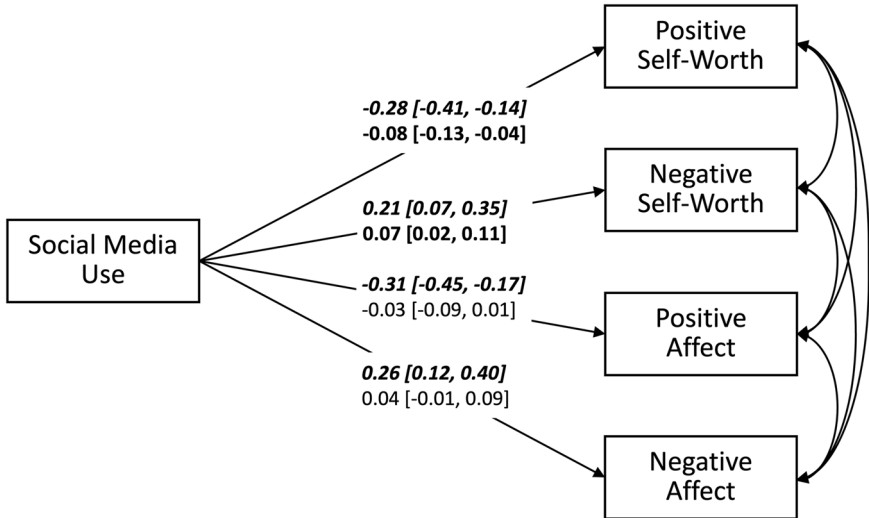

**Fig. 1 Predicting Subjective Well-Being by Social Media Use.** Schematic summary of a multilevel structural equation model predicting subjective well-being by social media use. $N = 200$ participants. Presented are standardized regression coefficients and their 95% credible interval. Upper lines written in italics refer to the between-person level and lower lines refer to the within-person level. Parameters whose 95% credible interval does not include zero and, hence, are interpreted as significant are highlighted in bold.

comparisons. On both the between- and within-person level, social media use positively predicted upward social comparisons (between: $\beta = 0.39$ [0.27, 0.51], $R^2 = 15.5\%$ [7.0%, 26.1%], within: $\beta = 0.09$ [0.05, 0.14], $R^2 = 5.6\%$ [3.7%, 7.8%]). Thus, children and young adolescents who used more Instagram, TikTok, and YouTube than others across the two weeks of assessment reported a more extreme impression that others had a better life, were happier, prettier, or more popular or had and did cooler things. Analogously, days with higher than usual social media use were days with a more extreme impression that others were better off.

Figure 3 depicts the results of the MSEM examining the associations between upward social comparisons and subjective well-being. On the between- and within-person level, upward social comparisons significantly predicted all four indicators of subjective well-being. Thus, children and young adolescents who on average engaged in more extreme upward social comparisons than others across the two weeks of assessment reported lower positive self-worth ($\beta = -0.61$ [$-0.69$, $-0.50$], $R^2 = 36.7\%$ [24.5%, 47.9%]) and positive affect ($\beta = -0.56$ [$-0.65$, $-0.43$], $R^2 = 30.9\%$ [18.8%, 42.5%]), and higher negative self-worth ($\beta = 0.68$ [0.59, 0.76], $R^2 = 45.9\%$ [34.4%, 57.1%]) and negative affect ($\beta = 0.67$ [0.57, 0.75], $R^2 = 44.2\%$ [31.9%, 55.5%]). Likewise, on days children and young adolescents had a more extreme impression that others were better off, they reported lower positive self-worth ($\beta = -0.26$ [$-0.31$, $-0.20$], $R^2 = 13.6\%$ [10.2%, 17.4%]) and positive affect ($\beta = -0.24$ [$-0.29$, $-0.18$], $R^2 = 14.9\%$ [11.4%, 18.8%]), and higher negative self-worth ($\beta = 0.27$ [0.22, 0.33], $R^2 = 13.6\%$ [10.5%, 17.0%]) and negative affect ($\beta = 0.27$ [0.22, 0.32], $R^2 = 14.9\%$ [11.7%, 18.6%]).

Figure 4 and Table 2 show the results of the multilevel mediation SEM. On the between-person level, the effects of social media use on upward social comparisons ($\beta = 0.39$ [0.25, 0.51]) and of upward social comparisons on the four indicators of subjective well-being were significant (positive self-worth: $\beta = -0.67$ [$-0.68$, $-0.46$], negative self-worth: $\beta = 0.69$ [0.58, 0.78], positive affect: $\beta = -0.48$ [$-0.60$, $-0.35$], and negative affect: $\beta = 0.62$ [0.50, 0.72]). Furthermore, all total and indirect effects of social media use on the four indicators of subjective well-being were significant (see Table 2). By contrast, the direct effects of social media use on the four indicators of subjective well-being were not significant (positive self-worth: $\beta = -0.05$

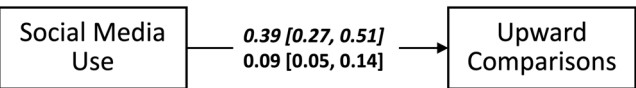

**Fig. 2 Predicting Upward Social Comparisons by Social Media Use.** Schematic summary of a multilevel structural equation model predicting upward social comparisons by social media use. $N = 200$ participants. Presented are standardized regression coefficients and their 95% credible interval. Upper lines written in italics refer to the between-person level and lower lines refer to the within-person level. Parameters whose 95% credible interval does not include zero and, hence, are interpreted as significant are highlighted in bold.

[$-0.18$, 0.08], negative self-worth: $\beta = -0.05$ [$-0.17$, 0.07], positive affect: $\beta = -0.13$ [$-0.27$, 0.00], and negative affect: $\beta = 0.04$ [$-0.09$, 0.17]). This indicates that the mediation model supported the assumption that the between-person effects of social media use on subjective well-being were fully mediated by upward social comparisons. Hence, children and young adolescents who used more social media than others reported more extreme upward social comparisons than others and these more extreme upward social comparisons were associated with reduced subjective well-being. $R^2$ on the between-person level was 15.4% [6.5%, 25.8%] for upward social comparisons, 35.7% [24.1%, 47.1%] for positive self-worth, 45.0% [32.9%, 56.3%] for negative self-worth, 30.4% [18.8%, 42.1%] for positive affect, and 41.1% [28.5%, 52.5%] for negative affect.

On the within-person level, the effects of social media use on upward social comparisons and of upward social comparisons on the four indicators of subjective well-being were significant (positive self-worth: $\beta = -0.24$ [$-0.30$, $-0.19$], negative self-worth: $\beta = 0.25$ [0.19, 0.31], positive affect: $\beta = -0.23$ [$-0.29$, $-0.17$], and negative affect: $\beta = 0.26$ [0.20, 0.31]). Furthermore, the indirect effects of social media use on positive and negative self-worth were significant, while the indirect effects on positive and negative affect were not (see Table 2). Only the total effect of social media use on positive self-worth was significant, but not the total effects of social media use on the other three indicators of subjective well-being (see Table 2). The direct effects of social media use on negative self-worth, positive affect, and negative

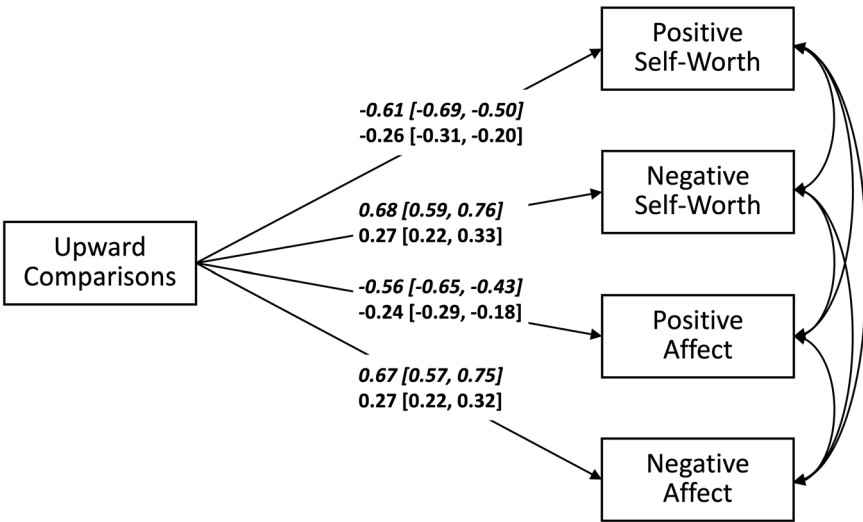

**Fig. 3 Predicting Subjective Well-Being by Upward Social Comparisons.** Schematic summary of a multilevel structural equation model predicting subjective well-being by upward social comparisons. $N = 200$ participants. Presented are standardized regression coefficients and their 95% credible interval. Upper lines written in italics refer to the between-person level and lower lines refer to the within-person level. Parameters whose 95% credible interval does not include zero and, hence, are interpreted as significant are highlighted in bold.

affect were not significant (negative self-worth: $\beta = 0.03$ [−0.02, 0.07], positive affect: $\beta = -0.01$ [−0.06, 0.03], and negative affect: $\beta = 0.01$ [−0.03, 0.05]). Only the direct effect of social media use on positive self-worth was significant ($\beta = -0.05$ [−0.10, −0.004]). These findings suggest that the within-person effects of social media use on self-worth were partly mediated by upward social comparisons. Hence, on days children and young adolescents used more social media than usually, they also experienced more extreme upward social comparisons than on other days and these more extreme upward social comparisons were linked to reduced subjective well-being on that day. $R^2$ on the within-person level was 6.5% [4.5%, 8.8%] for upward social comparisons, 17.8% [14.2%, 22.0%] for positive self-worth, 19.1% [15.5%, 22.8%] for negative self-worth, 21.3% [17.4%, 25.6%] for positive affect, and 21.4% [17.6%, 25.6%] for negative affect.

Furthermore, we tested whether the strengths of the associations between (1) social media use and each of the four subjective well-being indicators, (2) social media use and upward social comparisons, and (3) upward social comparisons and each of the four subjective well-being indicators varied depending on differences in person-level variables. The following person-level variables were examined as potential moderator variables: the child's sex, self-control failure in relation to social media use, the ability facet of social comparison orientation, and the opinion facet of social comparison orientation.

Of the 36 cross-level interaction effects, only one was found to be significant (see Supplementary Table 2). The negative effect of upward social comparisons on positive affect appeared to be weaker for children and young adolescents who reported that they often relied on the opinion of others (i.e., had a higher social comparison orientation regarding opinions) than for children and young adolescents who were less strongly oriented towards the opinion of others. However, we note that the significance of this effect should be considered critically due to the high number of cross-level interaction effects that we tested. Hence, we found no strong credible evidence that the strengths of the within-person associations between (1) social media use and each of the four subjective well-being indicators, (2) social media use and upward social comparisons, and (3) upward social comparisons and each of the four subjective well-being indicators could be explained by differences between youths in sex, self-control failure in relation

to social media use, comparison orientation regarding abilities, or comparison orientation regarding opinions.

## Discussion
This study showed that daily social media use was associated with decreased positive self-worth and increased negative self-worth in children and young adolescents. Additionally, upward social comparisons were consistently linked to reduced subjective well-being across various dimensions. Our findings further indicated that upward social comparisons (partially) mediated the effect of social media use on subjective well-being, both between individuals and within individuals over time. These findings suggest that youths' perceptions of others being better off than themselves may contribute to the heterogeneous results of previous research.

**Social media use and subjective well-being.** Within the scope of the present study, we developed several scales (e.g., assessing subjective social media use or upward social comparisons). For a further discussion of these new instruments see the Supplementary Discussion.

On average, we found social media use across the two weeks of assessments to be related to reduced subjective well-being. This indicates that children and young adolescents who used more Instagram, TikTok, and YouTube than others during the course of the study also reported to be less satisfied with themselves, more disappointed by or angry with themselves, to be less proud and to feel less good and content, and more unhappy, sad, and afraid than children and young adolescents who used social media less often. As the existing literature provides mixed results on the associations between social media use and well-being, the present findings are consistent with some[20–23] but not all previously published work[11–14,16–19]. Possible reasons for diverging results could be that the present study was conducted in individuals' natural everyday contexts. Hence, average social media use and average subjective well-being in our study referred to the arithmetic mean values of up to 14 assessments. Thereby, participants indicated their social media use and subjective well-being every evening retrospectively for the current day. In other studies, participants were instructed to estimate their average social media use or subjective well-being looking back over a

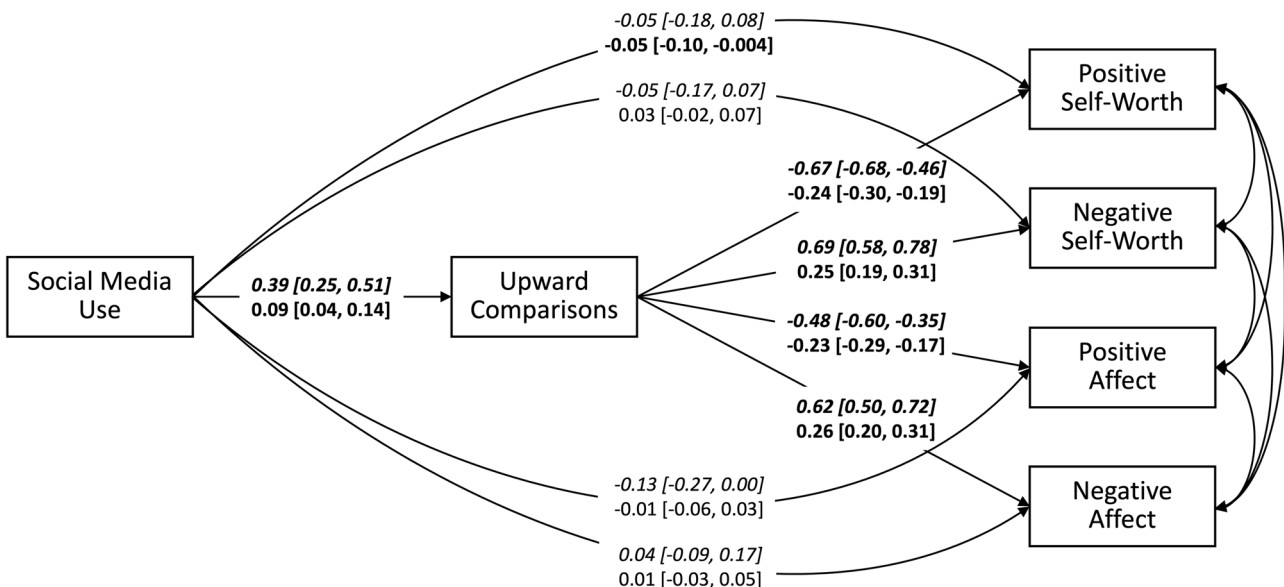

**Fig. 4 Upward Social Comparisons Mediating the Effects of Social Media Use on Subjective Well-Being.** Schematic representation of the results of a multilevel mediation model, with upward social comparisons mediating the effects of social media use on subjective well-being. $N = 200$ participants. Presented are standardized regression coefficients and their 95% credible interval. Upper lines written in italics refer to the between-person level and lower lines refer to the within-person level. Parameters whose 95% credible interval does not include zero and, hence, are interpreted as significant are highlighted in bold.

| Table 2 Total and Indirect Effects of the Multilevel Mediation Model. | | |
| --- | --- | --- |
| **Effect** | **Total Effect** | **Indirect Effect** |
| Within-person | | |
| Social media use → positive self-worth | **−0.16 [−0.27, −0.05]** | **−0.06 [−0.12, −0.01]** |
| Social media use → negative self-worth | 0.10 [−0.001, 0.21] | **0.05 [0.001, 0.10]** |
| Social media use → positive affect | −0.07 [−0.19, 0.06] | −0.03 [−0.09, 0.03] |
| Social media use → negative affect | 0.06 [−0.05, 0.16] | 0.04 [−0.01, 0.10] |
| Between-person | | |
| Social media use → positive self-worth | **−0.43 [−0.66, −0.21]** | **−0.35 [−0.51, −0.21]** |
| Social media use → negative self-worth | **0.26 [0.09, 0.43]** | **0.32 [0.20, 0.46]** |
| Social media use → positive affect | **−0.39 [−0.56, −0.22]** | **−0.23 [−0.34, −0.13]** |
| Social media use → negative affect | **0.25 [0.12, 0.38]** | **0.21 [0.13, 0.30]** |

Table shows parameter estimates and their 95% credible intervals in parentheses. Parameters whose 95% credible interval does not include zero and, hence, are interpreted as significant are highlighted in bold.

lifetime or a specific time period (e.g., several weeks up to one year). However, close-in-time questions answered under typical real-life conditions, as in our study, reduce the confounding influence of systematic biases in retrospective self-reports (e.g., recall biases) and increase the ecological validity of the study[74,92]. Yet, this difference in the assessments may have contributed to different findings. Another possible reason could be the time at which the current data were collected (April to June 2021). During these months, daily life was still strongly influenced by the COVID-19 pandemic. Recent research has shown that social media use of adults[93] and adolescents[94] increased during the pandemic. Moreover, social media use was found to be the only screen-media activity (besides television viewing, for instance) that was linked to worse mental health after the first lockdown in Swiss adolescents[94]. Furthermore, particularly young adults with high passive Facebook use and a strong social comparison orientation on Facebook reported a high level of psychological distress and low well-being during the pandemic in 2020[95]. Hence, there is evidence that individuals' engagement with social media during the COVID-19 pandemic was different than before[96]. Therefore, we cannot rule out the possibility that the

unique circumstances due to the pandemic influenced our findings, perhaps strengthening the associations between social media use and subjective well-being.

It has recently been claimed that the associations between social media use and well-being within individuals are "small to negligible"[35](p.5). The present study challenges this claim, as we (1) found substantial within-person fluctuations (i.e., day-to-day variations) of all items, (2) showed that these fluctuations were systematic (i.e., reliable) at the level of scales, and (3) found significant associations between social media use and self-worth. Specifically, we showed that on days children and young adolescents used more social media than usually, they were less satisfied and more dissatisfied with themselves, fitting in and extending the work by George et al.[36] who found social media use to be linked to same-day symptoms of inattention/hyperactivity and conduct disorder in adolescents. However, our result of significant within-person associations between social media use and self-worth stand in contrast to the results by Valkenburg et al.[39], who did not find a significant within-person effect of social media use on self-esteem. Yet, the authors assessed the use of Instagram, WhatsApp, and Snapchat; hence, only one platform

overlapped with the present study and the sample consisted of older participants than the current sample. Nonetheless, it is not clear why the current results diverge this much from the findings by Valkenburg et al.[39]. We did not find social media use to be associated with affective well-being on the same day. These findings are consistent with results reported by Beyens et al.[37] and Beyens et al.[38], who also did not find overall within-person effects of social media or Instagram use on affective well-being in adolescents and by Jensen et al.[18] who did not find daily technology entertainment to be related to mental health symptoms. Taking together, our findings indicate that the effects of social media use on subjective well-being depend on which indicator of subjective well-being is measured. In this context, our results suggest that self-worth, in particular, is a facet of subjective well-being that is associated with how much children and young adolescents use Instagram, TikTok, and YouTube in their everyday lives.

As mentioned before, we assessed participant's subjective evaluation of how much they had used social media. Different work has measured problematic use, typically with items referring to symptoms of addiction to social media (e.g., "During the past year, have you often found it difficult not to look at messages on social media when you were doing something else (e.g. school work)?"[97]) or intense use, which is operationalized by time spent on social media or by the frequency of engaging in different social media activities[4,23]. Thus, our measure is most closely related to intensity of social media use. There is (meta-analytic) evidence that problematic use exhibits stronger and more consistent associations with (reduced) well-being than intensity of use or time spent on social media[4,15,23,98]. It is therefore conceivable that the effects in our study would have been even stronger having assessed addiction-like symptoms of social media use instead of intensity of use. Future research should therefore examine the feasibility of assessing problematic social media use on a daily basis and compare the effects of both types of social media use on subjective well-being in youths.

**Social media use and upward social comparisons**. Extending prior work[23,54], we found that social media use and upward social comparisons were associated on an average and a day-to-day basis in youths. Hence, children and young adolescents who used more Instagram, TikTok, and YouTube across the two weeks of assessments experienced more extreme upward social comparisons than children and young adolescents who used social media less often. On a daily basis, this implies that days on which youths used more social media than usually were days on which they had a more extreme impression of others having a better life, doing cooler things, or being prettier and happier than themselves. Notably, this association was found to be significant although we did not distinguish among the type of use (i.e., active or passive) or the content of media consumption, and measured general (instead of media-related[23,79]) upward social comparisons. Our finding therefore imply that at least part of the general daily impression that others are better off is associated with daily social media use.

**Upward social comparisons and subjective well-being**. Our analyses further revealed that upward social comparisons were associated with both dimensions of self-worth and affective well-being on the between- and within-person level. This indicates that, on average and on a daily level, the impression of others being better off than oneself was linked to liking oneself less, feeling less proud and good about oneself, being more disappointed by and angry with oneself and to feeling more worthless, sad, and unhappy, and less good and content. This finding is in line with and extends existing between-person research with adolescents,

demonstrating that upward social comparisons on social media were associated with reduced well-being such as lower life satisfaction or more depressive symptoms[23,53,54,99].

**Upward social comparisons as a mediator**. Our between-person findings indicated a full mediation of the effects of social media use on subjective well-being via upward social comparisons. Hence, the direct effects of social media use on affective well-being and on self-worth were non-significant in a model including upward social comparisons alongside social media use. Thus, when accounting for their impression that others had a better life than they had, social media use was neither directly associated with children's positive or negative attitudes towards themselves nor with them feeling good or bad. However, social media use meaningfully predicted all four indicators of subjective well-being via upward social comparisons. This indicates that social media use was indirectly linked to reduced subjective well-being by strengthening users' impression of others being happier, more popular, and prettier or having cooler stuff or a better life. These findings are in line with between-person research with adults, showing upward social comparisons to mediate the relation among social media use and self-esteem or depressive symptoms[58–60] and with a study with 12- to 18-year-olds finding upward social comparisons to fully mediate to link between Qzone use and depression[11]. However, our results are not consistent with those reported by Boer et al.[23] who did not find evidence for upward social comparisons to mediate the longitudinal link between problematic social media use and depressive symptoms or life satisfaction[33]. Possible reasons for these inconsistencies could be the use of different measures of social media use and subjective well-being or differences in study design. Specifically, Boer et al.[23] conducted a longitudinal study and found problematic social media use to predict increases in upward social comparisons over time. However, they did not find these increases in upward social comparisons to predict decreases in mental health one year later, rejecting the expected mediation hypothesis. The authors supposed that "there may have been a mediating effect, but the measurements were possibly too far apart to observe it"[23](p.9), which is why they called for studies "using more intensive longitudinal data, such as daily measures of SMU [social media use] and mental health"[23](p.10). The present study followed this recommendation, suggesting that shorter time intervals between measurements are better suited to capture the mediating effect of upward social comparisons.

On a within-person level, we showed that social media use was no longer directly associated with negative self-worth when accounting for upward social comparisons. Yet, the indirect effect was significant, suggesting that daily social media use was associated with increased negative attitudes towards oneself on that day by increasing the impression that others had a better life, were prettier or more popular, for instance. For positive self-worth, we found that the effect of social media use was partially mediated by upward social comparisons. This indicates that on days children and young adolescents used more Instagram, TikTok, and YouTube than they usually did, they had a more extreme impression that others were better off than themselves, and this impression in turn was linked to decreased satisfaction and increased dissatisfaction with themselves. Furthermore, we found no statistically significant evidence for daily social media use to be linked to daily affective well-being in children and young adolescents, neither directly nor indirectly via upward social comparisons.

**Potentially moderating variables**. Our findings suggest that children who strongly rely on the opinion of others show a

weaker daily association between upward social comparisons and positive affect. However, considering the rather low reliability of the opinion scale and the high number of tests, the significance of this moderation effect has to be interpreted with caution and further research is to determine whether this finding can be replicated. Besides, we found no statistically significant evidence for children's and adolescents' sex, their self-control failure in relation to social media use, or their social comparison orientation on abilities to explain differences in the strengths of the associations examined in this work. Hence, contrary to previous research and our expectations, we found no statistically significant evidence for female participants to show stronger links than male participants. However, prior studies were mostly conducted with adults or adolescents; it is possible that sex differences are not yet observable at this young age and develop only in adolescence. In contrast to prior evidence[68–70], we further did not find that children with stronger self-control failure or with a higher social comparison orientation on abilities showed stronger or weaker links between the associations of interest.

Consequently, our results suggest that the associations between social media use, upward social comparisons, and subjective well-being hold across a range of person-level characteristics in children and young adolescents.

**Limitations**. Our study has several limitations. First, we used a convenience sample of children and young adolescents in Germany. Although invitation letters were sent out to all types of schools, primarily principals of the academic tier of secondary school forwarded our emails to the families of the target classes. Therefore, the sample was positively selected (i.e., high education and high income), limiting the generalizability of findings.

Second, there might be differences in the strength and direction of the relations between different social media platforms and well-being[75]. In our study, however, we concentrated on platforms that focus on visual content and we aggregated usage data of these platforms, in accordance with recommendations[76]. Nonetheless, future research should aim at systematically determining whether the effects of different social media platforms on upward social comparisons and subjective well-being are comparable in size and direction.

Third, we did not differentiate between active social media use (i.e., creating content, e.g., posting photos or videos) and passive social media use (e.g., consuming content). Future work should examine the extent to which children as young as ten to 14 years of age already engage in active social media use and, further, whether active and passive use differentially relate to upward social comparisons and subjective well-being.

Fourth, we developed our items assessing upward social comparisons based on previous work[23]. However, it may be argued that the items do not exactly tap the construct of engaging in upward social comparisons, but rather the impression or feeling that results from it. Therefore, it would be interesting to replicate our study with a different measure including items such as "Today, I compared myself to those who seemed to have a better life than me" and "Today, I compared myself to those who seemed to be prettier than me", and to compare such a measure to the present scale.

Fifth, our study relies on a correlational design, which is why the evidence is limited in terms of determining a causal order. For the indirect effects in the mediation model to be meaningfully interpreted in terms of a causal mediation model, there are two conditions that have to be met. First, a temporal sequence of constructs has to be assumed, and second, there must be no unobserved third variables responsible for the observed associations. Drawing such inferences from observational data is difficult

and we cannot rule out the possibility that there were other relevant common causes of our observed associations. For instance, social interactions in children's real lives could represent such a variable leading to an overestimation of the indirect effect: When, on certain days, children and young adolescents notice that peers look great at school, have the latest stuff and are popular, this plausibly induces upward social comparisons and, likewise, may reduce individual's self-worth.

And sixth, prior studies suggest that the associations among social media use, upward social comparisons, and subjective well-being might be reciprocal instead of one-directional[13,23,27,54,100,101]. For instance, it is also possible that elevated negative affect induces upward social comparisons, in that individuals have the feeling that everyone else is doing better than them. To deepen our understanding of the assumed complex interplay between social media use, upward social comparisons, and subjective well-being, future studies should examine the potential reciprocal relations on the within-day and across-day levels in children's and young adolescents' everyday lives.

**Reporting summary**. Further information on research design is available in the Nature Portfolio Reporting Summary linked to this article.

## Data availability
The data that support the findings of this study are openly available in the OSF at: https://osf.io/cs9um/.

## Code availability
Analysis code necessary to reproduce the results reported in this work are available in the Open Science Framework at https://osf.io/cs9um/.

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

## Acknowledgements

This study was funded by the Center for Research on Individual Development and Adaptive Education of Children at Risk (IDeA), Frankfurt am Main, Germany. We would like to thank Michaela Menstell and Anna-Maria Grimm for their valuable support in realizing the zEbra project.

## Author contributions

A.I.: conceptualization, funding acquisition, data collection, methodology, statistical analyses, and writing – original draft preparation; F.S.: project supervision and writing – review & editing. All authors read and approved the final version of the article.

## Funding

## Competing interests

The authors declare no competing interests.
