## [Peer Review File · Communications Psychology]

28th Mar 23

Dear Dr Irmer,

Thank you for your patience during the peer-review process. Your manuscript titled "The Daily Reminder That Others are Better off: The Role of Upward Comparisons in the Link Between Children's Everyday Social Media Use and Well-Being" has now been seen by 3 reviewers, and I include their comments at the end of this message. They find your work of interest, but raised some important points. We are interested in the possibility of publishing your study in *Communications Psychology*, but would like to consider your responses to these concerns and assess a revised manuscript before we make a final decision on publication.

We therefore invite you to revise and resubmit your manuscript, along with a point-by-point response to the reviewers. Please highlight all changes in the manuscript text file.

Reviewers #1 and #2 provided full length reports, while Reviewer #3 was contacted to provide feedback only on two issues on which the reports were in disagreement. This is why their feedback appears shorter.

Editorially, we consider it important that you address the reviewer's concerns regarding the measurement of social comparison. Please provide more detail on the measurement including greater specificity on what is being measured and how it affects your interpretation of the findings. Reviewers 2 and 3 also note the need to provide stronger rationale and support for your choice of moderator variables and age-range. As suggested by Reviewer 2, please make sure to incorporate Midgley et al., 2021 into your literature review and discussion. Last, we consider it important that you further discuss the limitations of your methods and findings.

Please avoid any novelty claims (describing the work as "novel", "first", "unprecedented", etc.), and the use of causal or mechanistic language to describe correlational data.

Please use the following link to submit your revised manuscript, point-by-point response to the referees' comments (which should be in a separate document to any cover letter) and the completed checklist:
[link redacted]

Please do not hesitate to contact me if you have any questions or would like to discuss these revisions further. We look forward to seeing the revised manuscript and thank you for the opportunity to review your work.

Best regards,

Jennifer Bellingtier

Jennifer Bellingtier, PhD
Senior Editor
Communications Psychology

EDITORIAL POLICIES AND FORMATTING

Editorial Policy: [Policy requirements](https://www.nature.com/documents/nr-editorial-policy-checklist.pdf) (Download the link to your computer as a PDF.)

Furthermore, please align your manuscript with our format requirements, which are summarized on the following checklist:

[Communications Psychology formatting checklist](https://www.nature.com/documents/commsj-style-formatting-checklist-review-perspective.pdf)

and also in our style and formatting guide [Communications Psychology formatting guide](https://www.nature.com/documents/commspsychol-style-formatting-guide-accept.pdf) .

* **CODE AVAILABILITY:** All Communications Psychology manuscripts must include a section titled "Code Availability" at the end of the methods section. In the event of publication, we require that the custom analysis code supporting your conclusions is made available in a publicly accessible repository; at publication, we ask you to choose a repository that provides a DOI for the code; the

link to the repository and the DOI will need to be included in the Code Availability statement. Publication as Supplementary Information will not suffice. We ask you to prepare code at this stage, to avoid delays later on in the process.

*** DATA AVAILABILITY:**

All Communications Psychology manuscripts must include a section titled "Data Availability" at the end of the Methods section or main text (if no Methods). More information on this policy, is available at <http://www.nature.com/authors/policies/data/data-availability-statements-data-citations.pdf>.

At a minimum the Data availability statement must explain how the data can be obtained and whether there are any restrictions on data sharing. Communications Psychology strongly endorses open sharing of data. If you do make your data openly available, please include in the statement:

We recommend submitting the data to discipline-specific, community-recognized repositories, where possible and a list of recommended repositories is provided at <http://www.nature.com/sdata/policies/repositories>.

If a community resource is unavailable, data can be submitted to generalist repositories such as [figshare](https://figshare.com/) or [Dryad Digital Repository](http://datadryad.org/). Please provide a unique identifier for the data (for example a DOI or a permanent URL) in the data availability statement, if possible. If the repository does not provide identifiers, we encourage authors to supply the search terms that will return the data. For data that have been obtained from publicly available sources, please provide a URL and the specific data product name in the data availability statement. Data with a DOI should be further cited in the methods reference section.

REVIEWERS' EXPERTISE:

Reviewer #1 Social comparison, experience sampling
Reviewer #2 Development, media use
Reviewer #3 Social comparison, media use

REVIEWERS' COMMENTS:

Reviewer #1 (Remarks to the Author):

In this manuscript, “The daily reminder that others are better off: The role of upward comparisons in the link between children’s everyday social media use and well-being,” the authors examine the between- and within-person associations between social media use, upward comparisons, and well-being (self-evaluations and affect) in children and adolescents using a 14-day daily diary design. Although the authors address an important question, examine this question in an important sample, use an ambitious design, and have done a great job explaining their analytical strategy, I have a number of concerns about the manuscript with regards to its contribution to the literature and some methodological issues.

1. Overall, the contribution of the manuscript to the literature is not as large as it could. Many of the findings presented in this manuscript have been tested and found in older samples (first-year undergraduate students and adults) using superior methods by Midgley et al. (2021). For example, in the present study, the authors measured upward comparisons in general and cannot be certain that social media use triggered these upward comparisons. Indeed, the authors even note that a plausible third variable could be social interactions with peers, which could lead to an overestimation in upward comparisons. This is consistent with past experience sampling studies showing that young adults compare themselves to their friends most often in daily life and often make upward comparisons in daily life (Wheeler & Miyake, 1992; Midgley et al., 2021). In contrast, Midgley et al. (2021) used an experience sampling design that asked participants whether they made a social comparison, the direction of the comparison, and in what context they made the comparison, which allowed them to directly link social media use to social comparisons and look at whether comparisons made on social media differed from comparisons made in other contexts. Midgley et al. (2021) also examine these associations at both the within-person and between-person levels in multiple studies; this is one of the key contributions that the authors claim they make in their current study. Furthermore, the current study relies on a correlational design; thus, the evidence is limited in terms of determining a causal order. The authors acknowledge that they do not have strong evidence for the proposed causal order in their study and that other causal orders are possible. The authors also do not test other potential causal orders. In contrast, Midgley et al. (2021) demonstrated that there is a bidirectional effect between self-esteem and upward comparisons on social media, such that social media use leads to more frequent upward comparisons which decreases self-evaluations, and lower self-esteem predicts more extreme upward comparisons (i.e., comparisons in which there is a larger discrepancy between the self and the superior other), which predicts greater decreases in self-evaluations. Moreover, Midgley et al. (2021) used an experimental design to show that social media use (vs. not using social media) does predict a greater likelihood of making an upward comparison and that in turn predicts decreased self-evaluations. This is the same mediation model that the authors present in the current manuscript. Consequently, the evidence presented in the current study is weaker than existing evidence. Although I strongly agree that it is important to examine these effects in younger samples, it is unclear why what Midgley et al. (2021) found would not also be true for younger samples. In fact, the authors suggest that these effects should be larger in younger samples because younger individuals have difficulty determining whether information on the Internet is valid relative to older peers. Given that existing research has found that these associations exist at both the between- and within-person levels, the present study is a replication of this work in a younger sample with less-than-ideal method, limiting the present study’s contribution to the literature.

Midgley, C., Thai, S., Lockwood, P., Kovacheff, C. & Page-Gould, E. (2021). When every day is a high school reunion: Social media comparisons and self-esteem. *Journal of Personality and Social Psychology*, 121, 285-307.

2. The authors identify a few potential moderators but do not explain their rationale for why these specific moderators would moderate the effects or why these moderators would be of interest to examine. Of note, given extensive media coverage regarding how social media use is particularly harmful to girls, it is surprising that this gender difference was not found in the data and not addressed by the authors at all. Furthermore, the authors then noted that the moderators they chose may not have been well-suited for testing these questions due to various reasons (e.g., low reliability, not tapping the relevant aspects of problematic social media use).

3. It is also unclear why the authors examined the specific age range that they did. Is this sample particularly vulnerable to the negative effects of social media? Are they undergoing important transitions that would be negatively affected by social media use? The current justification that these effects have not been examined in younger samples is insufficient and more justification is needed to understand the contributions of this research.

4. The authors did not describe the scale used for the upward comparison items, making it difficult to interpret the findings. The overall low means suggest that upward comparisons were relatively infrequent. Moreover, it is unclear what the authors mean by stronger upward comparisons. Do they mean these upward comparisons are more impactful? Do they mean that participants viewed a greater discrepancy between themselves and the other people? Do they mean the participants made more comparisons?

5. Another problem associated with using a global measure of upward comparisons is that they may not accurately reflect the degree to which people were engaging in actual comparison activity (Gerber et al. 2018). For example, when individuals feel worse (i.e., more negative affect), they may think that everyone else is doing better than them without making any comparisons at all. The items used by the authors in the current study and the correlational design allow this explanation to be a plausible, alternative explanation. Given this alternative explanation, the data presented in this manuscript do not provide strong evidence for their conclusions.

Reviewer #2 (Remarks to the Author):

Review of: COMMSPSYCHOL-23-0026-T

Thank you for the opportunity to review this manuscript, detailing the between- and within-subjects effects of daily social media use on aspects related to child and adolescent well-being. I have some comments that I hope are helpful and constructive to the authors. In short, though, I found this manuscript to be clear and well-written, the design and analysis important and well-conducted, and the findings have the opportunity to add to the literature in this research area.

1. In the introduction, it would be helpful to add a bit more on what you mean by the 'heterogeneity of studies' – I knew what the authors meant as I do research in this area, but I think one line or two might make this more clear for all readers.

2. On page 5, what do the authors mean by 'stable differences'?

3. Overall, I felt that the part on potential moderators to be very underdeveloped. This first shows up in the literature review when the authors mention that 'potential moderators' would be considered – but the reader does not find out which variables will be considered until the method, and even then, there is never an argument as to why these variables should be considered.

4. I think the authors did a good job of being clear when they are citing research on adolescents and when they are citing research on young adults or adults, but I would ask them to check the entire manuscript for this. It is fine to cite research on young adults, but given some of the developmental arguments they are making, it would be helpful to be clear about when the authors are citing research from these populations.
5. The authors could do more in the front end to clarify if they are conceptualizing and/or operationalizing social comparison at the state and trait level. Given that it is a mediator, I assumed it would be at the state level, but the authors then also measure it at the trait level as a moderator. In addition, it would be helpful to be clear whether the cited literature in this area conceptualized and operationalized it as a state or trait variable.
6. The authors write at multiple points that research 'in children's and adolescents' everyday lives is scarce' – what does this mean, exactly, and why is it important? I was not clear on this.
7. Can the authors provide the exact wording for the affective well-being measure?
8. I do not think this is a fatal flaw, and I'd like to see the authors respond to this, but given my concern about social comparison as a state or trait variable, how do the authors see this variable as operationalized as being state in nature? And given the design of the study, it is a little concerning that the adolescent is responding to these all at the end of the day. Given the authors are linking social media use with social comparison (and measuring both only at the end of the day), I would think the authors would have wanted to clarify that the social comparison was happening when on social media – otherwise, an adolescent could have socially compared themselves quite a bit at school, but in the analysis, it would look like their social media use were the antecedent of that social comparison, and not some event at school. I know the authors respond to this somewhat in the limitations section, but I think they could do a bit more to explicate how this choice might influence their findings pertaining to this variable.
9. Self-control failure and social media disorder seem like they would be at least moderately correlated, which made me wonder why both were selected as moderators, and in general, how these variables were selected to be moderators.
10. I was surprised that the mean for the social media use variables was fairly low – what do the authors make of this?
11. I realized that the authors refer to the variable as 'smartphone social media use' – but I am not sure this is quite right. The participants completed the surveys on a smartphone, but they could use social media on any device, right?
12. I think that section 4.1 could probably move to online supplementary files – while important, I did not immediately see what this added to the main papers and findings.
13. I think one big thing missing from discussion in the paper is COVID. Again, I do not think this is a fatal flaw, but these data were collected during the COVID pandemic, in a time when adolescents lives were still at least somewhat upended by the pandemic. Given that the authors make a point that other studies show inconsistent relations (but this study shows fairly clear relations) one explanation could be a function of the context of COVID. In other words, social media use might be linked to poorer well-being because of the unique stressors of the pandemic. I think it would be worth the authors discussing this further in the discussion section.
14. I felt like the paragraph in the discussion on active and passive social media use was not very well integrated (I do not think the authors had discussed this distinction to this point). I also felt it was a bit too speculative, so I would recommend cutting this.
15. At multiple points, the authors write that they go beyond previous findings by examining younger adolescents. Is there a developmental reason to expect that the relations of these variables among 10-14-year-olds would differ from those of 16-18-year-olds, for example? If so, this would be important to add. If not, I think the authors could spend a bit more less time making this distinction.

16. Finally, the authors frame the discussion of the moderators as being careful about focusing on the one significant interaction. I would advise to frame it more as -- these results are pretty consistent! I think that's the more noteworthy finding here. And if the authors focus more on why and how they selected these moderating variables, then I think this finding (or lack thereof in terms of significant interactions) will matter even more.

Reviewer #3 (Remarks to the Author):

There are not many studies done on social comparison using a daily diary approach, and this study adds to the existing literature such as Midgley et al., (2021).

The global measure of upward comparisons might better be described as measuring how people feel after the comparison rather than how much they compare to those who are better off. The scale could be better framed as measuring negative social comparison vs. upward social comparison.

The measurement of social media use is undifferentiated (as opposed to looking into active or passive social media use), which limits what we can learn from the findings.

Communications Psychology

April 28, 2023

Dear Reviewers,

Thank you very much for your evaluation of our manuscript and the interesting and helpful comments. In addressing your suggestions and concerns, we believe the manuscript has improved considerably.

For you to be able to follow the changes we made, we used the track changes feature of Microsoft Word and uploaded this file “Manuscript” as “Revised Manuscript – Marked Up”. Additionally, for better readability, we uploaded a file “Article_not marked up” as “Article file”. In our responses we inserted the new or adjusted paragraphs and included the page number (referring to the revised version with mark ups) to indicate precisely where changes were made. In doing the revision, we noticed an error in calculating the standard deviations of the three items assessing social media use. We corrected this in Table 1.

In the following, we describe point by point how we responded to your specific concerns:

Responses to Reviewer 1’s comments:

R1.0 In this manuscript, “The daily reminder that others are better off: The role of upward comparisons in the link between children’s everyday social media use and well-being,” the authors examine the between- and within-person associations between social media use, upward comparisons, and well-being (self-evaluations and affect) in children and adolescents using a 14-day daily diary design. Although the authors address an important question, examine this question in an important sample, use an ambitious design, and have done a great job explaining their analytical strategy, I have a number of concerns about the manuscript with regards to its contribution to the literature and some methodological issues.

RESPONSE: Thank you for the positive feedback on our study and the interesting suggestions that helped us to improve our manuscript.

R1.1 1. Overall, the contribution of the manuscript to the literature is not as large as it could. Many of the findings presented in this manuscript have been tested and found in older samples (first-year undergraduate students and adults) using superior methods by Midgley et al. (2021). For example, in the present study, the authors measured upward comparisons in general and cannot be certain that social media use triggered these upward comparisons. Indeed, the authors even note that a plausible third variable could be social interactions with peers, which could lead to an overestimation in upward comparisons. This is consistent with past experience sampling studies showing that young adults compare themselves to their friends most often in daily life and often make upward comparisons in daily life (Wheeler & Miyake, 1992; Midgley et al., 2021). In contrast, Midgley et al. (2021) used an experience sampling design that asked participants whether they made a social comparison, the direction of the comparison, and in what context they made the comparison, which allowed them to directly link social media use to social comparisons and look at whether comparisons made on social media differed from comparisons made in other contexts. Midgley et al. (2021) also examine these associations at both the within-person and between-person levels in multiple studies; this is one of the key contributions that the authors claim they make in their current study. Furthermore, the current study relies on a correlational design; thus, the evidence is limited in terms of determining a causal order. The authors

acknowledge that they do not have strong evidence for the proposed causal order in their study and that other causal orders are possible. The authors also do not test other potential causal orders. In contrast, Midgley et al. (2021) demonstrated that there is a bidirectional effect between self-esteem and upward comparisons on social media, such that social media use leads to more frequent upward comparisons which decreases self-evaluations, and lower self-esteem predicts more extreme upward comparisons (i.e., comparisons in which there is a larger discrepancy between the self and the superior other), which predicts greater decreases in self-evaluations. Moreover, Midgley et al. (2021) used an experimental design to show that social media use (vs. not using social media) does predict a greater likelihood of making an upward comparison and that in turn predicts decreased self-evaluations. This is the same mediation model that the authors present in the current manuscript. Consequently, the evidence presented in the current study is weaker than existing evidence. Although I strongly agree that it is important to examine these effects in younger samples, it is unclear why what Midgley et al. (2021) found would not also be true for younger samples. In fact, the authors suggest that these effects should be larger in younger samples because younger individuals have difficulty determining whether information on the Internet is valid relative to older peers. Given that existing research has found that these associations exist at both the between- and within-person levels, the present study is a replication of this work in a younger sample with less-than-ideal method, limiting the present study's contribution to the literature.

Midgley, C., Thai, S., Lockwood, P., Kovacheff, C. & Page-Gould, E. (2021). When every day is a high school reunion: Social media comparisons and self-esteem. *Journal of Personality and Social Psychology*, 121, 285-307.

RESPONSE: We thank the Reviewer for providing us with this very interesting study that we are happy to integrate in our manuscript. We extracted the specific concerns of the Reviewer and respond to them point by point in the following.

(1) For example, in the present study, the authors measured upward comparisons in general and cannot be certain that social media use triggered these upward comparisons.

We agree that we assessed upward social comparisons as the general impression that others have a better life, are prettier or happier, or do/have cooler stuff (see also our response to R1.5). The alternative would have been to directly refer to social media as did Boer et al (2021, p. 4), for instance:

“Respondents indicated, when viewing their peers’ messages, photos, or movies on social network sites, how often they thought ‘He or she does more fun things than I do’, ‘He or she has more friends than I do’ ...”. We were, however, interested in whether social media use is related to the overall impression that others are better off. This represents an extension of previous research and is why we measured global upward social comparisons and then, in the analyses, predicted this feeling by social media use. In our within-person analyses, we analyze whether variables (i.e., social media use and upward social comparisons) fluctuate simultaneously across time. Therefore, we were able to identify that on days children and young adolescents used more social media than usually, they also reported stronger upward social comparisons. Hence, at least part of the daily impression that others are better off seem to be associated with daily social media use – and to investigate this was one aim of the present study. We apologize for not having presented that clearly in the manuscript. We added this in the section “The Present Study” (pp. 14-15) and shortened respective paragraph in the Methods section:

“We assessed upward social comparisons as the general impression that others have a better life (e.g., are prettier, more popular, and happier or do and have cooler stuff). Notably, upward social comparisons scales used in other studies directly referred to social media (i.e., “Respondents indicated, when viewing their peers’ messages, photos, or movies on social network sites, how often they thought ‘He or she does more fun things than I do’”, Boer et al., 2021, p. 4; “When I read news feeds (or see

others' photos), I often think that others are having a better life than me", Lee, 2014, p. 256). One aim of the present study was to extend previous research by examining whether days with higher social media use were days on which children and young adolescents had a stronger impression of others having a better life, in general."

We also discuss on this further in the section 4.2 (p. 38):

"Furthermore, we deliberately measured general upward social comparisons, as opposed to different work that has assessed upward social comparisons triggered by social media use (see Boer et al., 2021; Lee, 2014). Hence, finding that on days children and young adolescents used more social media than usually, they also reported stronger upward social comparisons implies that at least part of the general daily impression that others are better off seem to be associated with daily social media use."

However, we acknowledge that upward social comparisons might have been induced by third variables in the present study. We now elaborate on this further in the Limitations (pp. 43-44):

"And fifth, prior studies suggest that the associations among social media use, upward social comparisons, and subjective well-being might be reciprocal instead of one-directional (Boer et al., 2021; Frison & Eggermont, 2016a, 2017; Midgley et al., 2021; Rousseau et al., 2017; Shin et al., 2022). For instance, it is also possible that elevated negative affect induces upward social comparisons, in that individuals have the feeling that everyone else is doing better than them. To deepen our understanding of the assumed complex interplay between social media use, upward social comparisons, and subjective well-being, future studies should aim at examining the potential reciprocal relations on the within-day and across-day levels in children's and young adolescents' everyday lives."

(2) Midgley et al. (2021) also examine these associations at both the within-person and between-person levels in multiple studies; this is one of the key contributions that the authors claim they make in their current study.

We thank the Reviewer for suggesting to include the study by Midgley et al. (2021) in our manuscript and to clearly present the additional value of our study. We incorporated the study at multiple sections within the manuscript; however, please note that within the entire manuscript, we focused on research with children and adolescents while limiting our references to studies with adults to a minimum.

In section 1.2 "Upward Social Comparisons" and 1.2.1 "Upward Social Comparisons as a Mediator" p. 7:

"Thus, they tend to perceive other's idealistic self-presentations as reality and automatically compare them with their own physical appearance, popularity, or leisure activities (Midgley et al., 2021)"

pp. 7-8:

"In line with this, recent research with undergraduates demonstrated that browsing social media primarily evoked upward social comparisons as compared to downward or lateral comparisons that are considered less harmful (Midgley et al., 2021). Furthermore, the authors showed that social comparisons that were made while using social media were more extremely upward than social comparisons occurring in different contexts. These upward social comparisons on social media were then found to be negatively associated with state self-esteem as well as life satisfaction after having used social media (Midgley et al., 2021)."

p. 9:

"For instance, Choi (2022) found that higher passive Instagram use was related to lower life satisfaction via upward social comparisons and Midgley et al. (2021) demonstrated that social media use was associated with feeling worse about oneself via upward social comparisons."

We also included the work by Midgley et al. (2021) in the Discussion:

p. 39:

“Building on previous work (e.g., Burnell et al., 2019; Midgley et al., 2021; Verduyn et al., 2020) and the results of the present study showing that both, social media use and upward social comparisons were associated with subjective well-being but also with one another, we examined upward social comparisons as a potential mechanism linking social media use to subjective well-being.”

p. 39:

“These findings are in line with between-person research with adults, showing upward social comparisons to mediate the relation among social media use and self-esteem or depressive symptoms (Midgley et al., 2021; Pang, 2021; Vogel et al., 2014; Wang et al., 2017).”

p. 43:

“Fourth, our study relies on a correlational design, which is why the evidence is limited in terms of determining a causal order. We decided which variables should serve as predictors and which as outcomes based on existing (experimental) research (e.g., Midgley et al. 2021); however, from our analyses, we cannot conclude whether upward social comparisons caused higher negative affect, for instance, or whether higher negative affect caused stronger upward social comparisons.”

p. 43:

“And fifth, prior studies suggest that the associations among social media use, upward social comparisons, and subjective well-being might be reciprocal instead of one-directional (Boer et al., 2021; Frison & Eggermont, 2016a, 2017; Midgley et al., 2021; Rousseau et al., 2017; Shin et al., 2022).”

One meaningful difference between the study by Midgley et al. (2021) and our study is that Midgley et al. examined undergraduates and middle-aged adults, whereas we examined children and young adolescents. In response to this comment and other comments, we now elaborate more clearly why we targeted the age group of ten- to 14-year-olds (see response to R1.3). For instance, our literature review revealed only two studies examining a potential mediation of the link between social media use and well-being via upward social comparisons in younger samples. Notably, these two studies revealed inconsistent results, as Niu et al. (2018) showed upward social comparisons to fully mediate the association between Qzone use and depression in 12- to 18-year-olds, whereas Boer et al. (2021) found no evidence for upward social comparisons to mediate the longitudinal link between problematic social media use and depressive symptoms or life satisfaction in 10- to 16-year-olds (see also Boer et al., 2022). Therefore, it is still unclear whether upward social comparisons represent a mechanism linking social media use to well-being in children and young adolescents. Moreover, both these studies (Boer et al., 2021; Niu et al., 2018) did not use a microlongitudinal or daily diary approach to examine the research questions. However, as elaborated in Molenaar (2004), associations found on the between-person level cannot simply be assumed to apply on a within-person level, likewise can results found in the laboratory not simply be transferred to real-life conditions (e.g., Ebner-Priemer et al., 2009). Due to these reasons we consider important that further research addresses the daily mechanisms linking social media use to well-being in children and adolescents.

Boer, M., Stevens, G. W., Finkenauer, C., Looze, M. E. de, & Van den Eijnden, R. J. (2021). Social media use intensity, social media use problems, and mental health among adolescents: Investigating directionality and mediating processes. *Computers in Human Behavior, 116*, 106645. <https://doi.org/10.1016/j.chb.2020.106645>

- Ebner-Priemer, U. W., Kubiak, T., & Pawlik, K. (2009). Ambulatory assessment. *European Psychologist, 14*(2), 95–97. <https://doi.org/10.1027/1016-9040.14.2.95>
- Molenaar, P. C. M. (2004). A manifesto on psychology as idiographic science: Bringing the person back into scientific psychology, this time forever. *Measurement: Interdisciplinary Research & Perspective, 2*(4), 201–218. https://doi.org/10.1207/s15366359mea0204_1
- Niu, G.-F., Luo, Y.-J., Sun, X.-J., Zhou, Z.-K., Yu, F., Yang, S.-L., & Zhao, L. (2018). Qzone use and depression among Chinese adolescents: A moderated mediation model. *Journal of Affective Disorders, 231*, 58–62. <https://doi.org/10.1016/j.jad.2018.01.013>

(3) Furthermore, the current study relies on a correlational design; thus, the evidence is limited in terms of determining a causal order. [...] The authors also do not test other potential causal orders.

We agree that our analyses do not allow to determine causality of effects. We decided which variables to be the predictor and which the outcome based on previous (experimental) research. However, we now acknowledge that causality cannot be inferred from our analyses in more detail in the Limitations (p. 43, see below). We also refer to potential bi-directional effects in the Limitations (pp. 43-44, see below):

“Fourth, our study relies on a correlational design, which is why the evidence is limited in terms of determining a causal order. We decided which variables should serve as predictors and which as outcomes based on existing (experimental) research (e.g., Midgley et al. 2021); however, from our analyses, we cannot conclude whether upward social comparisons caused higher negative affect, for instance, or whether higher negative affect caused stronger upward social comparisons. For the indirect effects in the mediation model to be meaningfully interpreted in terms of a causal mediation model, there are two conditions that have to be met for the indirect effects in the mediation model to be meaningfully interpreted in terms of a causal mediation model. First, a temporal sequence of constructs has to be assumed, and second, there must be no unobserved third variables responsible for the observed associations. Drawing such inferences from observational data is difficult and we cannot rule out the possibility that there were other relevant common causes of our observed associations. For instance, social interactions in children’s real lives could represent such a variable leading to an overestimation of the indirect effect: When, on certain days, children and young adolescents notice that peers look great at school, have the latest stuff and brand clothes, that they are popular and admired, this plausibly induces upward social comparisons and, likewise, may reduce individual’s self-worth.”

“And fifth, prior studies suggest that the associations among social media use, upward social comparisons, and subjective well-being might be reciprocal instead of one-directional (Boer et al., 2021; Frison & Eggermont, 2016a, 2017; Midgley et al., 2021; Rousseau et al., 2017; Shin et al., 2022). For instance, it is also possible that elevated negative affect induces upward social comparisons, in that individuals have the feeling that everyone else is doing better than them. To deepen our understanding of the assumed complex interplay between social media use, upward social comparisons, and subjective well-being, future studies should aim at examining the potential reciprocal relations on the within-day and across-day levels in children’s and young adolescents’ everyday lives.”

(4) This is the same mediation model that the authors present in the current manuscript. Consequently, the evidence presented in the current study is weaker than existing evidence. Although I strongly agree that it is important to examine these effects in younger samples, it is unclear why what Midgley et al. (2021) found would not also be true for younger samples. [...] Given that existing research has found that these associations exist at both the between- and within-person levels, the present study is a replication of this work in a younger sample with less-than-ideal method, limiting the present study’s contribution to the literature.

It is true that we examined the same mediation model as Midgley et al. (2021). We also acknowledge that presenting four studies in a paper denotes a significant contribution to the literature. However, we hope that we now comprehensibly presented our motivation to examine the relations in a younger sample. We agree that we would not have expected the significant associations found in Midgley et al. (2021) to be insignificant in this younger sample; yet, we believe that empirical evidence is needed to support (or challenge) respective assumptions. Besides, in the face of the current replication crisis of the social sciences, we believe that replications are of value and strongly needed to strengthen (or challenge) existing findings. This is particularly true for the present topic (social media use and well-being), as previous research has revealed highly inconsistent results. For instance, we found only two studies examining a potential mediation of the link between social media use and well-being via upward social comparisons in younger samples. Notably, these two studies revealed inconsistent results, as Niu et al. (2018) showed upward social comparisons to fully mediate the association between Qzone use and depression in 12- to 18-year-olds, whereas Boer et al. (2021) found no evidence for upward social comparisons to mediate the longitudinal link between problematic social media use and depressive symptoms or life satisfaction in 10- to 16-year-olds (see also Boer et al., 2022). Hence, further research, such as the present study, is needed to examine the associations of interest in children and adolescents.

R1.2 The authors identify a few potential moderators but do not explain their rationale for why these specific moderators would moderate the effects or why these moderators would be of interest to examine. Of note, given extensive media coverage regarding how social media use is particularly harmful to girls, it is surprising that this gender difference was not found in the data and not addressed by the authors at all. Furthermore, the authors then noted that the moderators they chose may not have been well-suited for testing these questions due to various reasons (e.g., low reliability, not tapping the relevant aspects of problematic social media use).

RESPONSE: We apologize that we did not comprehensibly justify our motivations for examining ten to 14-year-olds and for investigating the specific moderators. In this revision, we have elucidated the reasons more thoroughly in a new Section 1.3 “Potential Moderators” (pp. 9-11) and also extended the paragraph in the Discussion on this matter, addressing the insignificant moderation of gender (p. 42). Please note that in response to another comment (see response to R2.9), we now do not examine problematic social media use as a potential moderator anymore. Furthermore, in response to comment R2.16, we now highlight more clearly that having found only one significant moderation also points to highly consistent results (see paragraph below).

In the Theoretical Background (pp. 9-11):

“1.3 Potential Moderators

One aim of the present study was to identify person-level variables that moderate the associations between (1) social media use and subjective well-being, (2) social media use and upward social comparisons, or (3) upward social comparisons and subjective well-being. Based on previous research, we examined the following potential moderator variables: gender, self-control failure regarding social media use, and social comparison orientation.

Previous research has pointed to gender differences in the associations of interest in the present study. For instance, Nesi and Prinstein (2015) found that female adolescents showed a stronger link between technology-based social comparison and feedback-seeking with depressive symptoms than male adolescents. Furthermore, the intensity of social media use at age ten predicted well-being in adolescence for females, but not for males (Booker et al., 2018). Therefore, we included gender as a potential moderator variable, expecting girls to be more susceptible to the effects of social media and upward social comparisons than boys.

Besides gender, prior studies with adults also motivated us to examine self-control failure with regard to social media use. Considering problematic social media use as a continuum ranging from mild problems to limit social media use to extremely problematic (i.e., addiction-like) social media use, the construct of self-control failure falls within the lower side of the continuum. Hence, it assesses the degree to which individuals use social media platforms although it stands in conflict with other goals as well as the degree to which individuals use their time efficiently or delay other important duties due to social media (Du et al., 2018). Self-control failure related to social media use can be considered a predictive factor of later social media addiction (Li et al., 2014). Previous research has shown that high-intense users exhibit stronger associations between social media use and well-being. For instance, Boer et al. (2021) showed that problematic social media use was related to depressive symptoms and (negatively) to life satisfaction, but that intensity of social media use was unrelated to these variables in ten- to 16-year-olds. Moreover, work that has reviewed existing evidence with individuals of all ages concludes that self-control serves as a significant moderator of the effects of media use on well-being (Hofmann et al., 2016; Reinecke et al., 2022). Therefore, we decided to explore whether social media self-control failure moderates the associations of interest in the present study. Furthermore, there is evidence showing that individuals significantly differ in their general tendency to engage in social comparisons and that such differences moderate the effects of social media on well-being (De Vries et al., 2018; Kleemans et al., 2018; Vogel et al., 2015; Yang, 2016). In an experimental study with girls aged 14 to 18 years, those with a stronger tendency to compare themselves to others were more negatively affected by viewing manipulated Instagram posts (i.e., reported lower body satisfaction) than those with a weaker tendency to engage in social comparisons (Kleemans et al., 2018). Similarly, Vogel et al. (2015) conducted an experimental study and found that undergraduates with a strong tendency to compare themselves to others were more negatively affected by viewing others' Facebook profiles (i.e., reported lower self-esteem, lower affective well-being, and poorer self-perceptions) than undergraduates with a weaker tendency to compare themselves to others. Based on these studies that were mostly conducted with (young) adults, we investigated the general tendency to engage in social comparisons as another potential moderator in our study.”

And in the Discussion (p. 42):

“Hence, contrary to previous research and our expectations, we did not find that girls showed stronger links than boys or that children with stronger self-control failure or with stronger social comparison orientation on abilities showed stronger links than children with weaker self-control failure or with weaker social comparison orientation on abilities. However, prior studies were mostly conducted with adults or adolescents, whereas we examined children aged ten to 14 years. It is possible that gender differences, for instance, are not yet observable at this young age and develop only in adolescence. Nonetheless, our results suggest that the associations between social media use, upward social comparisons, and subjective well-being hold across a range of person-level characteristics in children and young adolescents.”

- Booker, C. L., Kelly, Y. J., & Sacker, A. (2018). Gender differences in the associations between age trends of social media interaction and well-being among 10-15 year olds in the UK. *BMC Public Health*, 18(1), 1-12. <https://doi.org/10.1186/s12889-018-5220-4>
- Hofmann, W., Reinecke, L., & Meier, A. (2016). Of sweet temptations and bitter aftertaste: Self-control as a moderator of the effects of media use on well-being. In *The Routledge handbook of media use and well-being* (pp. 211-222). Routledge.
- Kleemans, M., Daalmans, S., Carbaat, I., & Anschütz, D. (2018). Picture perfect: The direct effect of manipulated Instagram photos on body image in adolescent girls. *Media Psychology*, 21(1), 93-110. <https://doi.org/10.1080/15213269.2016.1257392>

- Li, C., Dang, J., Zhang, X., Zhang, Q., & Guo, J. (2014). Internet addiction among Chinese adolescents: The effect of parental behavior and self-control. *Computers in Human Behavior, 41*, 1–7. <https://doi.org/10.1016/j.chb.2014.09.001>
- Nesi, J., & Prinstein, M. J. (2015). Using social media for social comparison and feedback-seeking: Gender and popularity moderate associations with depressive symptoms. *Journal of Abnormal Child Psychology, 43*(8), 1427–1438. <https://doi.org/10.1007/s10802-015-0020-0>
- Reinecke, L., Gilbert, A., & Eden, A. (2022). Self-regulation as a key boundary condition in the relationship between social media use and well-being. *Current Opinion in Psychology, 45*, 101296. <https://doi.org/10.1016/j.copsyc.2021.12.008>
- Yang, C. C. (2016). Instagram use, loneliness, and social comparison orientation: Interact and browse on social media, but don't compare. *Cyberpsychology, Behavior, and Social Networking, 19*(12), 703-708. <http://dx.doi.org/10.1089/cyber.2016.0201>.

R1.3 It is also unclear why the authors examined the specific age range that they did. Is this sample particularly vulnerable to the negative effects of social media? Are they undergoing important transitions that would be negatively affected by social media use? The current justification that these effects have not been examined in younger samples is insufficient and more justification is needed to understand the contributions of this research.

RESPONSE: Thank you for pointing out that our motivation for examining ten to 14-year-olds has not been comprehensibly justified. In this revision, we have elucidated the reasons more thoroughly in Section 1.4 “The Present Study” (pp. 11-12), while incorporating three more studies on this matter: “We targeted this age group due to several reasons. First, research has shown that by the age of ten, children mostly start using social media on smartphones (Hasebrink et al., 2019; Krebs & Rynkowski, 2019). However, it is also starting in middle childhood that individuals develop a sense of self and form their identities (Eccles, 1999). As this process is usually accompanied by social comparisons, these also become more and more important during this time (Eccles, 1999; Erikson, 1968). Social media platforms offer endless opportunities to compare to others, as users can follow and watch friends or strangers around the world, provoking comparisons with multiple persons in many aspects (Verduyn et al., 2020). Thus, the second reason we targeted children and adolescents ages ten to 14 was to examine the associations between social media use, upward social comparisons, and subjective well-being (e.g., self-esteem) in an era characterized by figuring out one's worth based on comparisons with others. Third, previous research has suggested children to be particularly susceptible to the (negative) effects of media use (Booker et al., 2018; Liu et al., 2015), which has also been described as “developmental susceptibility” (Valkenburg & Peter, 2013, p. 227). For instance, high social media use in children (i.e., ten-year-olds) has been shown to have implications for subjective well-being in adolescence, especially for girls (Booker et al., 2018). Furthermore, in a meta-analysis, Liu et al. (2015) found that higher screen time was associated with a higher risk for developing depression in children from ten to 14 years of age, but not for those older than 14 years. Hence, as children appear to be especially vulnerable to the harmful effects of social media use on well-being and healthy development, research examining the associations in this age group’s everyday lives in more detail is essential to develop tailored and effective prevention and intervention measures.”

- Booker, C. L., Kelly, Y. J., & Sacker, A. (2018). Gender differences in the associations between age trends of social media interaction and well-being among 10-15 year olds in the UK. *BMC Public Health, 18*(1), 1-12. <https://doi.org/10.1186/s12889-018-5220-4>
- Liu, M., Wu, L., & Yao, S. (2016). Dose–response association of screen time-based sedentary behaviour in children and adolescents and depression: A meta-analysis of observational studies. *British Journal of Sports Medicine, 50*(20), 1252-1258. <http://dx.doi.org/10.1136/bjsports-2015-095084>

R1.4 The authors did not describe the scale used for the upward comparison items, making it difficult to interpret the findings. The overall low means suggest that upward comparisons were relatively infrequent. Moreover, it is unclear what the authors mean by stronger upward comparisons. Do they mean these upward comparisons are more impactful? Do they mean that participants viewed a greater discrepancy between themselves and the other people? Do they mean the participants made more comparisons?

RESPONSE: We thank the Reviewer for pointing out that information on response scales was not provided for the upward social comparison scale. We added this together with further information on what higher scores (i.e., stronger upward comparisons) indicate on p. 17:

“The items were answered on a 5-point scale (1 = “not at all true” to 5 = “completely true”). Hence, higher scores on this scale (i.e., referred to as stronger upward social comparisons in the following) indicate that children and young adolescents had the impression that others had a better life than themselves or were happier, prettier, or more popular, that is, they perceived a higher discrepancy between themselves and others.”

The wording of each item can also be found in Table 1, showing descriptive statistics of all items. We added this information in section 2.3.1 (p. 16):

“The wording and descriptive statistics of all items of daily measures are presented in Table 1.”

We agree that the mean levels of upward social comparisons were rather low. However, the intraclass correlations and intraindividual standard deviations show that there was variability between and within children, which is an important prerequisite for our analyses. Within-person analyses consider couplings between variables, that is, we analyzed whether or not variables fluctuated or varied simultaneously. For such analyses, the level of responses (how much participants agreed with the items) is of little relevance.

R1.5 Another problem associated with using a global measure of upward comparisons is that they may not accurately reflect the degree to which people were engaging in actual comparison activity (Gerber et al. 2018). For example, when individuals feel worse (i.e., more negative affect), they may think that everyone else is doing better than them without making any comparisons at all. The items used by the authors in the current study and the correlational design allow this explanation to be a plausible, alternative explanation. Given this alternative explanation, the data presented in this manuscript do not provide strong evidence for their conclusions.

RESPONSE: We understand the point raised by the Reviewer and agree that our correlational design does not allow to determine the causality of associations.

Yet, when planning our study and developing the items, we purposely decided to assess upward social comparisons as a global impression (see also our response to R1.1). The alternative would have been to directly refer to social media as did Boer et al (2021, p. 4), for instance: “Respondents indicated, when viewing their peers’ messages, photos, or movies on social network sites, how often they thought ‘He or she does more fun things than I do’, ‘He or she has more friends than I do’ ...”. We were, however, interested in whether social media use is related to the overall impression that others are better off. This represents an extension of previous research and is why we measured global upward social comparisons and then, in the analyses, predicted this feeling by social media use. In our within-person analyses, we analyze whether variables (i.e., social media use and upward social comparisons) fluctuate simultaneously across time. Therefore, we were able to identify that on days children and young adolescents used more social media than usually, they also reported stronger upward social

comparisons. Hence, at least part of the daily impression that others are better off seem to be associated with daily social media use. Nonetheless, we agree with the Reviewer that we cannot rule out that negative affect might also induce upward social comparisons. We now elaborate on this further in the Discussion:

p. 38:

“Furthermore, we deliberately measured general upward social comparisons, as opposed to different work that has assessed upward social comparisons triggered by social media use (see Boer et al., 2021; Lee, 2014). Hence, finding that on days children and young adolescents used more social media than usually, they also reported stronger upward social comparisons implies that at least part of the general daily impression that others are better off seem to be associated with daily social media use.”

pp. 43-44:

“And fifth, prior studies suggest that the associations among social media use, upward social comparisons, and subjective well-being might be reciprocal instead of one-directional (Boer et al., 2021; Frison & Eggermont, 2016a, 2017; Midgley et al., 2021; Rousseau et al., 2017; Shin et al., 2022). For instance, it is also possible that elevated negative affect induces upward social comparisons, in that individuals have the feeling that everyone else is doing better than them. To deepen our understanding of the assumed complex interplay between social media use, upward social comparisons, and subjective well-being, future studies should aim at examining the potential reciprocal relations on the within-day and across-day levels in children’s and young adolescents’ everyday lives.”

Responses to Reviewer 2’s comments:

R2.0 Thank you for the opportunity to review this manuscript, detailing the between- and within-subjects effects of daily social media use on aspects related to child and adolescent well-being. I have some comments that I hope are helpful and constructive to the authors. In short, though, I found this manuscript to be clear and well-written, the design and analysis important and well-conducted, and the findings have the opportunity to add to the literature in this research area.

RESPONSE: We thank the Reviewer for the positive evaluation of our work and the helpful feedback and suggestions.

R2.1 In the introduction, it would be helpful to add a bit more on what you mean by the ‘heterogeneity of studies’ – I knew what the authors meant as I do research in this area, but I think one line or two might make this more clear for all readers.

RESPONSE: We agree with the Reviewer and added further information on p. 3:

“The authors highlighted the heterogeneity of studies, summarizing that some meta-analyses found small to moderate associations of social media use with higher levels of adolescent ill-being, while another meta-analyses found social media use to be unrelated to ill-being and others found weak associations with higher levels of well-being. Therefore, they called for within-person studies as well as mediation and moderator analyses that allow to shed light on the potential complex associations of social media use and well-being in youths.”

R2.2 On page 5, what do the authors mean by ‘stable differences’?

RESPONSE: In this paragraph, we aim at contrasting between- and within-person findings. Between-person findings are based on differences between individuals, whereas within-person findings refer to fluctuations of variables across time. For example, between-person differences in social media use reflect the degree to which individuals differ in their average levels of social media use whereas within-person variations in social media use reflect the degree to which individuals' levels of social media use

deviate from their average level across situations. However, we agree that “stable” is somewhat misleading and therefore deleted this word while adding further clarifying information on p. 5: “In fact, it has recently been argued that the associations between social media use and well-being are mainly driven by differences between individuals (i.e., differences in the average levels of variables), while the effects within individuals across time are “small to negligible” (Johannes et al., 2022, p. 5).”

R2.3 Overall, I felt that the part on potential moderators to be very underdeveloped. This first shows up in the literature review when the authors mention that ‘potential moderators’ would be considered – but the reader does not find out which variables will be considered until the method, and even then, there is never an argument as to why these variables should be considered.

RESPONSE: We apologize that we did not comprehensibly justify our motivations for examining the specific moderators. In this revision, we have elucidated the reasons more thoroughly in a new Section 1.3 “Potential Moderators” (pp. 9-11):

“One aim of the present study was to identify person-level variables that moderate the associations between (1) social media use and subjective well-being, (2) social media use and upward social comparisons, or (3) upward social comparisons and subjective well-being. Based on previous research, we examined the following potential moderator variables: gender, self-control failure regarding social media use, and social comparison orientation.

Previous research has pointed to gender differences in the associations of interest in the present study. For instance, Nesi and Prinstein (2015) found that female adolescents showed a stronger link between technology-based social comparison and feedback-seeking with depressive symptoms than male adolescents. Furthermore, the intensity of social media use at age ten predicted well-being in adolescence for females, but not for males (Booker et al., 2018). Therefore, we included gender as a potential moderator variable, expecting girls to be more susceptible to the effects of social media and upward social comparisons than boys.

Besides gender, prior studies with adults also motivated us to examine self-control failure with regard to social media use. Considering problematic social media use as a continuum ranging from mild problems to limit social media use to extremely problematic (i.e., addiction-like) social media use, the construct of self-control failure falls within the lower side of the continuum. Hence, it assesses the degree to which individuals use social media platforms although it stands in conflict with other goals as well as the degree to which individuals use their time efficiently or delay other important duties due to social media (Du et al., 2018). Self-control failure related to social media use can be considered a predictive factor of later social media addiction (Li et al., 2014). Previous research has shown that high-intense users exhibit stronger associations between social media use and well-being. For instance, Boer et al. (2021) showed that problematic social media use was related to depressive symptoms and (negatively) to life satisfaction, but that intensity of social media use was unrelated to these variables in ten- to 16-year-olds. Moreover, work that has reviewed existing evidence with individuals of all ages concludes that self-control serves as a significant moderator of the effects of media use on well-being (Hofmann et al., 2016; Reinecke et al., 2022). Therefore, we decided to explore whether social media self-control failure moderates the associations of interest in the present study.

Furthermore, there is evidence showing that individuals significantly differ in their general tendency to engage in social comparisons and that such differences moderate the effects of social media on well-being (De Vries et al., 2018; Kleemans et al., 2018; Vogel et al., 2015; Yang, 2016). In an experimental study with girls aged 14 to 18 years, those with a stronger tendency to compare themselves to others were more negatively affected by viewing manipulated Instagram posts (i.e., reported lower body satisfaction) than those with a weaker tendency to engage in social comparisons (Kleemans et al., 2018). Similarly, Vogel et al. (2015) conducted an experimental study and found that undergraduates with a strong tendency to compare themselves to others were more negatively affected by viewing others’ Facebook profiles (i.e., reported lower self-esteem, lower affective well-being, and poorer self-

perceptions) than undergraduates with a weaker tendency to compare themselves to others. Based on these studies that were mostly conducted with (young) adults, we investigated the general tendency to engage in social comparisons as another potential moderator in our study.”

- Booker, C. L., Kelly, Y. J., & Sacker, A. (2018). Gender differences in the associations between age trends of social media interaction and well-being among 10-15 year olds in the UK. *BMC Public Health*, *18*(1), 1-12. <https://doi.org/10.1186/s12889-018-5220-4>
- Hofmann, W., Reinecke, L., & Meier, A. (2016). Of sweet temptations and bitter aftertaste: Self-control as a moderator of the effects of media use on well-being. In *The Routledge handbook of media use and well-being* (pp. 211-222). Routledge.
- Kleemans, M., Daalmans, S., Carbaat, I., & Anschütz, D. (2018). Picture perfect: The direct effect of manipulated Instagram photos on body image in adolescent girls. *Media Psychology*, *21*(1), 93-110. <https://doi.org/10.1080/15213269.2016.1257392>
- Li, C., Dang, J., Zhang, X., Zhang, Q., & Guo, J. (2014). Internet addiction among Chinese adolescents: The effect of parental behavior and self-control. *Computers in Human Behavior*, *41*, 1–7. <https://doi.org/10.1016/j.chb.2014.09.001>
- Nesi, J., & Prinstein, M. J. (2015). Using social media for social comparison and feedback-seeking: Gender and popularity moderate associations with depressive symptoms. *Journal of Abnormal Child Psychology*, *43*(8), 1427–1438. <https://doi.org/10.1007/s10802-015-0020-0>
- Reinecke, L., Gilbert, A., & Eden, A. (2022). Self-regulation as a key boundary condition in the relationship between social media use and well-being. *Current Opinion in Psychology*, *45*, 101296. <https://doi.org/10.1016/j.copsyc.2021.12.008>
- Yang, C. C. (2016). Instagram use, loneliness, and social comparison orientation: Interact and browse on social media, but don't compare. *Cyberpsychology, Behavior, and Social Networking*, *19*(12), 703-708. <http://dx.doi.org/10.1089/cyber.2016.0201>.

R2.4 I think the authors did a good job of being clear when they are citing research on adolescents and when they are citing research on young adults or adults, but I would ask them to check the entire manuscript for this. It is fine to cite research on young adults, but given some of the developmental arguments they are making, it would be helpful to be clear about when the authors are citing research from these populations.

RESPONSE: We carefully checked the manuscript and added respective information on populations if necessary. We also made sure that all new paragraphs include this information. You may find three examples in the following (see pp. 5-6, p. 8, and p. 11). Thereby, we replaced a study with young adults by another study with adolescents (Kleemans et al., 2018).

“For instance, George et al. (2018) found social media use to be linked to same-day symptoms of inattention/hyperactivity and conduct disorder in 11- to 15-year-olds across a 30-day period of daily assessments.”

“Empirical research with adolescents supported this finding, showing that upward social comparisons on social media are associated with diminished well-being (e.g., Boer et al., 2021; Nesi & Prinstein, 2015; Rousseau et al., 2017; see also Meier & Johnson, 2022, for a cross-age review).”

“In an experimental study with girls aged 14 to 18 years, those with a stronger tendency to compare themselves to others were more negatively affected by viewing manipulated Instagram posts (i.e., reported lower body satisfaction) than those with a weaker tendency to engage in social comparisons (Kleemans et al., 2018).”

R2.5 The authors could do more in the front end to clarify if they are conceptualizing and/or operationalizing social comparison at the state and trait level. Given that it is a mediator, I assumed it would be at the state level, but the authors then also measure it at the trait level as a moderator. In addition, it would be helpful to be clear whether the cited literature in this area conceptualized and operationalized it as a state or trait variable.

RESPONSE: Thank you for raising this interesting point. A lot of research has been devoted on the question of whether traits, that is, rather time-invariant, stable characteristics of individuals or time-varying, unstable characteristics of situations are more important when predicting behaviors (e.g., Donnellan; Funder, 2008; Kenrick & Funder, 1988). Studies have shown that most constructs contain both, rather stable trait parts and fluctuating state parts. Thereby, constructs differ in the degree to which they contain the one of other component; hence, they could be classified into more “trait-like” (e.g., intelligence) and more “state-like” variables (e.g., mood). Interestingly, personality, which had been assumed to be a highly stable characteristic for a long time, has also been shown to comprise stable parts (see, e.g., special issue “New approaches towards conceptualizing and assessing personality” in the *European Journal of Personality*, 2020). Coming back to your question, we conceptualized upward social comparisons as a state when measuring it on a daily basis, using items such as “Today, I had the feeling that others have a better life than me” or “Today, I had the feeling that others are more popular than I am”. With this scale, we assessed every day how much children and young adolescents had the feeling or impression that others were doing better than they are. We added more information on this on p. 17:

“The items were answered on a 5-point scale (1 = “not at all true” to 5 = “completely true”). Hence, higher scores on this scale (i.e., referred to as stronger upward social comparisons in the following) indicate that children and young adolescents had the impression that others had a better life than themselves or were happier, prettier, or more popular, that is, they perceived a higher discrepancy between themselves and others.”

However, we also assessed the trait component of social comparison using the German version of the well-established Iowa-Netherlands Comparison Orientation Measure (INCOM) by Gibbons & Buunk (1999). This scale assesses individual’s general tendency to compare with others, sample items are “I do not often compare myself to others (r)” or “I always like to know what others in a similar situation would do”. Hence, compared to the state upward social comparison scale, the INCOM does not include a direction of comparison (i.e., upward or downward).

We clarified this when presenting the person-level measures on p. 18:

“The scale assesses individual’s general tendency to compare themselves with others.”

In the section on upward social comparisons in our manuscript, we mainly cited cross-sectional (or longitudinal, but not experience sampling) studies, which is why social comparison had been conceptualized and operationalized as a trait variable in the literature we referred to. As such, many studies (e.g., Kleemans et al., 2018; Vogel et al., 2015) used the Iowa-Netherlands Comparison Orientation Measure (INCOM) by Gibbons & Buunk (1999) that we had also used as the trait measure. One exception is the study by Midgley et al. (2021) who also conceptualized upward social comparisons as a state measure. In their experience sampling study, undergraduates were randomly prompted six times per day. In the surveys, they were then asked whether they had made a social comparison since the last survey and, in case participants indicated they had made a comparison, there were further questions (e.g., to whom they had compared). Hence, the authors also conceptualized upward social comparisons as a state, fluctuating over time.

- Donnellan, M. B., Lucas, R. E., & Fleeson, W. (2009). Editorial: Introduction to personality and assessment at age 40: Reflections on the legacy of the person-situation debate and the future of person-situation integration. *Journal of Research in Personality, 43*, 117–119.
- Funder, D. C. (2008). Persons, situations, and person-situation interactions. In O. P. John, R. Robins, & L. Pervin (Eds.), *Handbook of personality* (3rd ed, pp. 568–580). New York, NY: Guilford Press
- Gibbons, F., & Buunk, B. (1999). Individual differences in social comparison: Development of a scale of social comparison orientation. *Journal of Personality and Social Psychology, 76*(1), 129–142. <https://doi.org/10.1037//0022-3514.76.1.129>
- Kenrick, D. T., & Funder, D. C. (1988). Profiting from controversy: Lessons from the person-situation debate. *The American Psychologist, 43*, 23–34.
- Kleemans, M., Daalmans, S., Carbaat, I., & Anschütz, D. (2018). Picture perfect: The direct effect of manipulated Instagram photos on body image in adolescent girls. *Media Psychology, 21*(1), 93–110. <https://doi.org/10.1080/15213269.2016.1257392>
- Vogel, E. A., Rose, J. P., Okdie, B. M., Eckles, K., & Franz, B. (2015). Who compares and despairs? The effect of social comparison orientation on social media use and its outcomes. *Personality and Individual Differences, 86*, 249–256. <https://doi.org/10.1016/j.paid.2015.06.026>

R2.6 The authors write at multiple points that research ‘in children’s and adolescents’ everyday lives is scarce’ – what does this mean, exactly, and why is it important? I was not clear on this.

RESPONSE: Measurement-intensive studies (e.g., daily diary studies as the current one) can be conducted in the laboratory or in the natural contexts of individuals’ everyday lives. Collecting data in individuals’ everyday lives as opposed to the laboratory is assigned high ecological validity, as it captures behaviors or feelings in spontaneous, natural environments. Thus, effects are studied under typical conditions in the real world, diminishing response biases resulting from unusual laboratory settings and allowing to account for contextual factors (Ebner-Priemer et al., 2009; Reis, 2012; Shiffman et al., 2008). Furthermore, measuring constructs repeatedly in natural settings further has advantages over traditional survey or retrospective methods (Trull & Ebner-Priemer, 2014), as participants respond to real-time questions (i.e., referring to the actual moment) or close-in-time questions (e.g., referring to the current day). This minimizes the confounding influence of systematic biases in retrospective self-reports (e.g., recall biases; Bolger & Laurenceau, 2013; Schwarz, 2012; see Shiffman et al., 2008, for a review). We added respective information on pp. 12-13:

“Collecting data in individuals’ daily lives provides high ecological validity, because behaviors and emotions are captured in the real world, and their associations are investigated under typical conditions and in natural contexts (Ebner-Priemer et al., 2009).”

- Bolger, N., & Laurenceau, J.-P. (2013). Intensive longitudinal methods: An introduction to diary and experience sampling research. *Methodology in the social sciences*. New York, NY: Guilford Press.
- Ebner-Priemer, U. W., Kubiak, T., & Pawlik, K. (2009). Ambulatory assessment. *European Psychologist, 14*, 95–97. <https://doi.org/10.1027/1016-9040.14.2.95>
- Reis, H. T. (2012). Why researchers should think “Real-world”: A conceptual rationale. In M. R. Mehl & T. S. Conner (Eds.), *Handbook of research methods for studying daily life* (pp. 3–21). New York, NY: Guilford Press.
- Schwarz, N. (2012). Why researchers should think “Real-time”: A cognitive rationale. In M. R. Mehl & T. S. Conner (Eds.), *Handbook of research methods for studying daily life* (pp. 22–42). New York, NY: Guilford Press.
- Shiffman, S., Stone, A. A., & Hufford, M. R. (2008). Ecological momentary assessment. *Annual Review of Clinical Psychology, 4*, 1–32. <https://doi.org/10.1146/annurev.clinpsy.3.022806.091415>

Trull, T. J., & Ebner-Priemer, U. (2014). The role of ambulatory assessment in psychological science. *Current Directions in Psychological Science*, 23, 466–470. <https://doi.org/10.1177/0963721414550706>

R2.7 Can the authors provide the exact wording for the affective well-being measure?

RESPONSE: Positive affect was measured with these items: “Today, I felt good / fantastic / content”. Negative affect was assessed with the following items: “Today, I felt unhappy / sad / miserable / afraid”. The exact wording and descriptive statistics of all variables can be found in Table 1 on pp. 22-23. The items were used in previous ambulatory assessment studies with children (Leonhardt et al., 2016; Schmidt et al., 2020; Schmidt et al., 2019) and adolescents (Limberger et al., under review).

- Leonhardt, A., Könen, T., Dirk, J., & Schmiedek, F. (2016). How differentiated do children experience affect? An investigation of the within- and between-person structure of children's affect. *Psychological Assessment*, 28(5), 575–585. <https://doi.org/10.1037/pas0000195>
- Limberger, M. F., Schmiedek, F., Santangelo, P., Reichert, M., Meyer-Lindenberg, A., Tost, H., & Ebner-Priemer, U. W. (under review). Assessing affect in adolescents with e-diaries: Model fit and reliability in multilevel confirmatory factor analyses of different factor models.
- Schmidt, A., Dirk, J., & Schmiedek, F. (2019). The importance of peer relatedness at school for affective well-being in children: Between- and within-person associations. *Social Development*, 28(4), 873–892. <https://doi.org/10.1111/sode.12379>
- Schmidt, A., Neubauer, A. B., Dirk, J., & Schmiedek, F. (2020). The bright and the dark side of peer relationships: Differential effects of relatedness satisfaction and frustration at school on affective well-being in children's daily lives. *Developmental Psychology*, 56(8), 1532–1546. <https://doi.org/10.1037/dev0000997>

R2.8 I do not think this is a fatal flaw, and I'd like to see the authors respond to this, but given my concern about social comparison as a state or trait variable, how do the authors see this variable as operationalized as being state in nature? And given the design of the study, it is a little concerning that the adolescent is responding to these all at the end of the day. Given the authors are linking social media use with social comparison (and measuring both only at the end of the day), I would think the authors would have wanted to clarify that the social comparison was happening when on social media – otherwise, an adolescent could have socially compared themselves quite a bit at school, but in the analysis, it would look like their social media use were the antecedent of that social comparison, and not some event at school. I know the authors respond to this somewhat in the limitations section, but I think they could do a bit more to explicate how this choice might influence their findings pertaining to this variable.

RESPONSE: Again, we thank the Reviewer for raising the interesting questions about social comparisons being state or trait in nature (see our response to R2.5). We agree with the Reviewer that it would have been nice to have multiple assessments of all variables per day. However, when planning our study, we had to weigh the benefit of multiple assessments against the burden of study participation. Verbeij et al. (2021) included retrospective surveys of time spent on social media in the previous week as well as daily experience sampling questions on the time spent social media in the previous hour. They found that the convergent validity of retrospective surveys and the experience sampling method were more or less equivalent. In the discussion, the authors emphasize that “researchers should therefore consider whether their research questions justify the potential physical burden (e.g., the high number of questionnaires) of an ESM design, given that our ESM estimates did not lead to a higher validity of adolescents’ average time estimates across the three-week ESM period than our retrospective surveys did” (p. 9). However, they add that multiple assessments are inevitable when studying within-

person associations. Therefore, we opted for a daily diary approach, which is less burdensome than an ESM study but allows the investigation of processes within participants.

We assessed upward social comparisons as the general impression that others have a better life, are prettier or happier, or do/have cooler stuff (see also our response to R1.5). The reviewers observed that we did not measure upward social comparisons directly referring to social media use as did Boer et al (2021, p. 4), for instance: “Respondents indicated, when viewing their peers’ messages, photos, or movies on social network sites, how often they thought ‘He or she does more fun things than I do’, ‘He or she has more friends than I do’ ...”. We were, however, interested in whether social media use is related to the overall impression that others are better off. This represents an extension of previous research and is why we measured global upward social comparisons and then, in the analyses, predicted this feeling by social media use. In our within-person analyses, we analyze whether variables (i.e., social media use and upward social comparisons) fluctuate simultaneously across time. Therefore, we were able to identify that on days children and young adolescents used more social media than usually, they also reported stronger upward social comparisons. Hence, at least part of the daily impression that others are better off seem to be associated with daily social media use – and to investigate this was one aim of the present study. We apologize for not having presented that clearly in the manuscript. We added this in the section “The Present Study” (pp. 14-15) and shortened respective paragraph in the Methods section:

“We assessed upward social comparisons as the general impression that others have a better life (e.g., are prettier, more popular, and happier or do and have cooler stuff). Notably, upward social comparisons scales used in other studies directly referred to social media (i.e., “Respondents indicated, when viewing their peers’ messages, photos, or movies on social network sites, how often they thought ‘He or she does more fun things than I do’”, Boer et al., 2021, p. 4; “When I read news feeds (or see others’ photos), I often think that others are having a better life than me”, Lee, 2014, p. 256). One aim of the present study was to extend previous research by examining whether days with higher social media use were days on which children and young adolescents had a stronger impression of others having a better life, in general.”

We also discuss on this further in section 4.2 (p. 38):

“Furthermore, we deliberately measured general upward social comparisons, as opposed to different work that has assessed upward social comparisons triggered by social media use (see Boer et al., 2021; Lee, 2014). Hence, finding that on days children and young adolescents used more social media than usually, they also reported stronger upward social comparisons implies that at least part of the general daily impression that others are better off seem to be associated with daily social media use.”

As Reviewer 2 already mentioned, we discussed the possibility that events at school triggered upward social comparisons in the Limitations. However, we now additionally acknowledge that upward social comparisons might have been induced by other third variables (i.e., negative affect) in the present study (pp. 43-44):

“And fifth, prior studies suggest that the associations among social media use, upward social comparisons, and subjective well-being might be reciprocal instead of one-directional (Boer et al., 2021; Frison & Eggermont, 2016a, 2017; Midgley et al., 2021; Rousseau et al., 2017; Shin et al., 2022). For instance, it is also possible that elevated negative affect induces upward social comparisons, in that individuals have the feeling that everyone else is doing better than them. To deepen our understanding of the assumed complex interplay between social media use, upward social comparisons, and subjective well-being, future studies should aim at examining the potential reciprocal relations on the within-day and across-day levels in children’s and young adolescents’ everyday lives.”

Verbeij, T., Pouwels, J. L., Beyens, I., & Valkenburg, P. M. (2021). The accuracy and validity of self-reported social media use measures among adolescents. *Computers in Human Behavior Reports*, 3, 100090. <https://doi.org/10.1016/j.chbr.2021.100090>

R2.9 Self-control failure and social media disorder seem like they would be at least moderately correlated, which made me wonder why both were selected as moderators, and in general, how these variables were selected to be moderators.

RESPONSE: We thank the Reviewer for this comment and agree that the measures of self-control failure and social media disorder are related. The researchers who developed the self-control failure scale also referred to this relation in detail (Du et al., 2018), pointing out that problematic social media use should be considered as a continuum ranging from mild self-control failures to extremely pathological (i.e., addiction-like) forms of social media use. Hence, typical social media disorder scales capture the more extreme forms of pathological use, while, according to the authors, there was a need for scales assessing lighter types of problematic social media use. Another reason that motivated them to develop the self-control failure scale was a low prevalence of addiction-like social media (or Internet) use (about 2-6 % of adult users; see Du et al., 2018). In their study, they showed that the scale correlated about .50 with other scales on addiction-like or problematic social media use, while these measures correlated about .80 with each other. Self-control failure can also be considered as a predictive factor of the development of Internet addiction (Li et al., 2014).

However, as we assumed the prevalence of addiction-like social media use to be even lower in children than in the general population (due to parental restrictions of smartphone use, for instance), we decided to disseminate the self-control failure scale besides the social media disorder scale. However, we agree with the Reviewer that the scales have similarities and, as there were no fundamental differences in the results we found for both potential moderator variables, we decided to remove the analyses on social media disorder. We now also incorporate a section explicating our motivation for examining the different moderator variables (see section 1.3 “Potential Moderators” on pp. 9-11).

Du, J., van Koningsbruggen, G. M., & Kerkhof, P. (2018). A brief measure of social media self-control failure. *Computers in Human Behavior*, 84, 68-75. <https://doi.org/10.1016/j.chb.2018.02.002>

Li, C., Dang, J., Zhang, X., Zhang, Q., & Guo, J. (2014). Internet addiction among Chinese adolescents: The effect of parental behavior and self-control. *Computers in Human Behavior*, 41, 1-7. <https://doi.org/10.1016/j.chb.2014.09.001>

R2.10 I was surprised that the mean for the social media use variables was fairly low – what do the authors make of this?

RESPONSE: Participants in our study were young compared to other studies. Therefore, it was not possible to estimate the amount of social media use at this young age before data collection. We agree that the mean level of Instagram use, in particular, was rather low. However, the intraclass correlations and intraindividual standard deviations show that there was variability between and within children, which is an important prerequisite for our analyses. Within-person analyses consider couplings between variables, that is, we analyzed whether or not variables fluctuate or vary simultaneously. For such analyses, the level of responses (how much participants agreed with an item) is of little relevance.

R2.11 I realized that the authors refer to the variable as ‘smartphone social media use’ – but I am not sure this is quite right. The participants completed the surveys on a smartphone, but they could use social media on any device, right?

RESPONSE: We apologize for not having made that clear within the manuscript. When participants were introduced into the study, they had been instructed to consider only their smartphone use of these platforms. We added this information on p. 16:

“At the beginning of the study, participants had been instructed to consider only their smartphone use of these platforms.”

R2.12 I think that section 4.1 could probably move to online supplementary files – while important, I did not immediately see what this added to the main papers and findings.

RESPONSE: We agree with the Reviewer and moved this section to the online supplement. We now refer to the section at the beginning of the new section 4.1 (p. 33):

“Within the scope of the present study, we developed several scales (i.e., assessing subjective social media use). For a further discussion of these new instruments see the Online Supplement S3.”

R2.13 I think one big thing missing from discussion in the paper is COVID. Again, I do not think this is a fatal flaw, but these data were collected during the COVID pandemic, in a time when adolescents lives were still at least somewhat upended by the pandemic. Given that the authors make a point that other studies show inconsistent relations (but this study shows fairly clear relations) one explanation could be a function of the context of COVID. In other words, social media use might be linked to poorer well-being because of the unique stressors of the pandemic. I think it would be worth the authors discussing this further in the discussion section.

RESPONSE: Thank you for raising this important point. We now discuss on this on pp. 34-35:

“Another possible reason could be the time at which the current data were collected (April to June 2021). During these months, daily life was still strongly influenced by the COVID-19 pandemic and related measures to contain the spread of the virus. Recent research has shown that social media use of adults (Gudino et al., 2022) and adolescents (Marciano et al., 2022) increased during the pandemic. Moreover, Marciano et al. (2022) demonstrated that social media use was the only screen-media activity (besides video gaming or television viewing, for instance) that was linked to worse mental health after the first lockdown in Swiss adolescents. Furthermore, Bonfanti et al. (2022) found that young adults with high passive Facebook use, a strong social comparison orientation on Facebook, and high fear of missing out reported a high level of psychological distress and low well-being during the pandemic in 2020. Hence, there is evidence that individuals’ engagement with social media during the COVID-19 pandemic was different than before (see also Turner & Ordonia, 2023). Therefore, we cannot rule out the possibility that the unique circumstances due to the pandemic influenced our findings, perhaps strengthening the associations between social media use and subjective well-being.”

Bonfanti, R. C., Salerno, L., Brugnera, A., & Lo Coco, G. (2022). A longitudinal investigation on problematic Facebook use, psychological distress and well-being during the second wave of COVID-19 pandemic. *Scientific Reports*, *12*(1), 21828.

<https://doi.org/10.1038/s41598-022-26281-0>

Gudiño, D., Fernández-Sánchez, M.J., Becerra-Traver, M.T., Sánchez, S. (2022). Social media and the pandemic: Consumption habits of the Spanish population before and during the COVID-19 lockdown. *Sustainability*, *14*, 5490. <https://doi.org/10.3390/su14095490>

Marciano, L., Viswanath, K., Morese, R., & Camerini, A. L. (2022). Screen time and adolescents' mental health before and after the COVID-19 lockdown in Switzerland: A natural experiment. *Frontiers in Psychiatry*, *13*, 981881. <https://doi.org/10.3389/fpsy.2022.981881>

Turner, M., & Ordonia, D. (2023). How COVID-19 changed self-presentation on instagram and its relation to user well-being. *Interacting with Computers*, Advance Online Publication.

<https://doi.org/10.1093/iwc/iwad013>

R2.14 I felt like the paragraph in the discussion on active and passive social media use was not very well integrated (I do not think the authors had discussed this distinction to this point). I also felt it was a bit too speculative, so I would recommend cutting this.

RESPONSE: We agree and therefore removed this paragraph. However, please note that Reviewer 3 mentioned the missing distinction between active and passive use in the present study as a limitation, which is why we have included this in respective section (p. 43).

“Third, we did not differentiate between active social media use (i.e., creating content, e.g., posting photos or videos) and passive social media use (e.g., consuming content) in this work. Future work should examine the extent to which children as young as ten to 14 years of age already engage in active social media use and, further, whether active and passive use differentially relate to upward social comparisons and subjective well-being.”

R2.15 At multiple points, the authors write that they go beyond previous findings by examining younger adolescents. Is there a developmental reason to expect that the relations of these variables among 10-14-year-olds would differ from those of 16-18-year-olds, for example? If so, this would be important to add. If not, I think the authors could spend a bit more less time making this distinction.

RESPONSE: We apologize for not having stated clearly why we examined the age group of ten to 14 years. We have added a paragraph on this in section 1.4 “The Present Study” (pp. 11-12):

“We targeted this age group due to several reasons. First, research has shown that by the age of ten, children mostly start using social media on smartphones (Hasebrink et al., 2019; Krebs & Rynkowski, 2019). However, it is also starting in middle childhood that individuals develop a sense of self and form their identities (Eccles, 1999). As this process is usually accompanied by social comparisons, these also become more and more important during this time (Eccles, 1999; Erikson, 1968). Social media platforms offer endless opportunities to compare to others, as users can follow and watch friends or strangers around the world, provoking comparisons with multiple persons in many aspects (Verduyn et al., 2020). Thus, the second reason we targeted children and adolescents ages ten to 14 was to examine the associations between social media use, upward social comparisons, and subjective well-being (e.g., self-esteem) in an era characterized by figuring out one's worth based on comparisons with others. Third, previous research has suggested children to be particularly susceptible to the (negative) effects of media use (Booker et al., 2018; Liu et al., 2015), which has also been described as “developmental susceptibility” (Valkenburg & Peter, 2013, p. 227). For instance, high social media use in children (i.e., ten-year-olds) has been shown to have implications for subjective well-being in adolescence, especially for girls (Booker et al., 2018). Furthermore, in a meta-analysis, Liu et al. (2015) found that higher screen time was associated with a higher risk for developing depression in children from ten to 14 years of age, but not for those older than 14 years. Hence, as children appear to be especially vulnerable to the harmful effects of social media use on well-being and healthy development, research examining the associations in this age group's everyday lives in more detail is essential to develop tailored and effective prevention and intervention measures.”

Booker, C. L., Kelly, Y. J., & Sacker, A. (2018). Gender differences in the associations between age trends of social media interaction and well-being among 10-15 year olds in the UK. *BMC Public Health*, 18(1), 1-12. <https://doi.org/10.1186/s12889-018-5220-4>

Liu, M., Wu, L., & Yao, S. (2016). Dose–response association of screen time-based sedentary behaviour in children and adolescents and depression: A meta-analysis of observational studies. *British Journal of Sports Medicine*, 50(20), 1252-1258. <http://dx.doi.org/10.1136/bjsports-2015-095084>

Valkenburg, P. M., & Peter, J. (2013). The differential susceptibility to media effects model. *Journal of Communication*, 63(2), 221-243. <https://doi.org/10.1111/jcom.12024>

R2.16 Finally, the authors frame the discussion of the moderators as being careful about focusing on the one significant interaction. I would advise to frame it more as -- these results are pretty

consistent! I think that's the more noteworthy finding here. And if the authors focus more on why and how they selected these moderating variables, then I think this finding (or lack thereof in terms of significant interactions) will matter even more.

RESPONSE: We thank the Reviewer for this comment and now highlight more clearly that having found only one significant interaction points to very consistent results (p. 42):

“Hence, contrary to previous research and our expectations, we did not find that girls showed stronger links than boys or that children with stronger self-control failure or with stronger social comparison orientation on abilities showed stronger links than children with weaker self-control failure or with weaker social comparison orientation on abilities. However, prior studies were mostly conducted with adults or adolescents, whereas we examined children aged ten to 14 years. It is possible that gender differences, for instance, are not yet observable at this young age and develop only in adolescence. Nonetheless, our results suggest that the associations between social media use, upward social comparisons, and subjective well-being hold across a range of person-level characteristics in children and young adolescents.”

Responses to Reviewer 3's comments:

R3.0 There are not many studies done on social comparison using a daily diary approach, and this study adds to the existing literature such as Midgley et al., (2021).

RESPONSE: We thank the Reviewer for valuing the contribution of our study to the literature.

R3.1 The global measure of upward comparisons might better be described as measuring how people feel after the comparison rather than how much they compare to those who are better off. The scale could be better framed as measuring negative social comparison vs. upward social comparison.

RESPONSE: Thank you for your suggestion on renaming the construct. We agree that it would also have been possible to assess how individuals felt after the comparison. This was done by Buunk et al. (2005), for instance, using the following two items: “How often do you feel good when you see that colleagues perform worse in their work than you do yourself?” and “How often do you feel bad when you see that colleagues perform better in their work than you do yourself?” (see p. 67). However, they called the construct “Affective consequences of social comparison”. Similarly, Midgley et al. (2021) used two items measuring how individuals felt after comparing: “After making this comparison, I felt better about myself” and “After making this comparison, I felt worse about myself” [reverse-scored] (see p. 289). The authors called this construct “Postcomparison self-evaluation”. By contrast, we did not ask children how they felt after the comparison but asked them whether they agreed to statements implying the comparison to a superior other. Thereby, we referred to previous work on “Upward (social) comparisons” (e.g., Boer et al., 2021; Schmuck et al., 2019; Vogel et al., 2014). This term is well-established, a definition is provided by Gerber et al. (2018): “An upward comparison is one in which the comparison standard is better off than the comparer” (p. 178; see also Collins et al., 1996). Boer et al. (2021) also assessed “Upward social comparisons” in adolescents and used items such as “He or she does more fun things than I do” or “He or she receives more ‘likes’ than me”. Our items were developed based on these items and, hence, are quite similar. Therefore, we tend to keep framing our scale as upward social comparisons.

Boer, M., Stevens, G. W., Finkenauer, C., Looze, M. E. de, & Van den Eijnden, R. J. (2021). Social media use intensity, social media use problems, and mental health among adolescents: Investigating directionality and mediating processes. *Computers in Human Behavior, 116*, 106645. <https://doi.org/10.1016/j.chb.2020.106645>

- Buunk, B. P., Zurriaga, R., Peiró, J. M., Nauta, A., & Gosalvez, I. (2005). Social comparisons at work as related to a cooperative social climate and to individual differences in social comparison orientation. *Applied Psychology, 54*(1), 61-80.
- Collins, R. L. (1996). For better or worse: The impact of upward social comparisons on self-evaluations. *Psychological Bulletin, 119*, 51– 69. <http://dx.doi.org/10.1037/0033-2909.119.1.51>
- Gerber, J. P., Wheeler, L., & Suls, J. (2018). A social comparison theory meta-analysis 60+ years on. *Psychological Bulletin, 144*(2), 177–197. <https://doi.org/10.1037/bul0000127>
- Midgley, C., Thai, S., Lockwood, P., Kovacheff, C. & Page-Gould, E. (2021). When every day is a high school reunion: Social media comparisons and self-esteem. *Journal of Personality and Social Psychology, 121*(2), 285-307. <http://dx.doi.org/10.1037/pspi0000336>
- Schmuck, D., Karsay, K., Matthes, J., & Stevic, A. (2019). “Looking up and feeling down”. The influence of mobile social networking site use on upward social comparison, self-esteem, and well-being of adult smartphone users. *Telematics and Informatics, 42*, 101240. <https://doi.org/10.1016/j.tele.2019.101240>
- Vogel, E. A., Rose, J. P., Roberts, L. R., & Eckles, K. (2014). Social comparison, social media, and self-esteem. *Psychology of Popular Media Culture, 3*(4), 206–222. <https://doi.org/10.1037/ppm0000047>

R3.2 The measurement of social media use is undifferentiated (as opposed to looking into active or passive social media use), which limits what we can learn from the findings.

RESPONSE: We thank the Reviewer for raising this point and added a paragraph to the Limitations on p. 43:

“Third, we did not differentiate between active social media use (i.e., creating content, e.g., posting photos or videos) and passive social media use (e.g., consuming content) in this work. Future work should examine the extent to which children as young as ten to 14 years of age already engage in active social media use and, further, whether active and passive use differentially relate to upward social comparisons and subjective well-being.”

References

- Bolger, N., & Laurenceau, J.-P. (2013). *Intensive longitudinal methods: An introduction to diary and experience sampling research. Methodology in the social sciences*. New York, NY: Guilford Press.
- Boer, M., Stevens, G. W., Finkenauer, C., Looze, M. E. de, & Van den Eijnden, R. J. (2021). Social media use intensity, social media use problems, and mental health among adolescents: Investigating directionality and mediating processes. *Computers in Human Behavior, 116*, 106645. <https://doi.org/10.1016/j.chb.2020.106645>
- Booker, C. L., Kelly, Y. J., & Sacker, A. (2018). Gender differences in the associations between age trends of social media interaction and well-being among 10-15 year olds in the UK. *BMC Public Health, 18*(1), 1-12. <https://doi.org/10.1186/s12889-018-5220-4>
- Buunk, B. P., Zurriaga, R., Peiró, J. M., Nauta, A., & Gosalvez, I. (2005). Social comparisons at work as related to a cooperative social climate and to individual differences in social comparison orientation. *Applied Psychology, 54*(1), 61-80.
- Collins, R. L. (1996). For better or worse: The impact of upward social comparisons on self-evaluations. *Psychological Bulletin, 119*, 51– 69. <http://dx.doi.org/10.1037/0033-2909.119.1.51>
- Ebner-Priemer, U. W., Kubiak, T., & Pawlik, K. (2009). Ambulatory assessment. *European Psychologist, 14*, 95–97. <https://doi.org/10.1027/1016-9040.14.2.95>
- Gerber, J. P., Wheeler, L., & Suls, J. (2018). A social comparison theory meta-analysis 60+ years on. *Psychological Bulletin, 144*(2), 177–197. <https://doi.org/10.1037/bul0000127>
- Hofmann, W., Reinecke, L., & Meier, A. (2016). Of sweet temptations and bitter aftertaste: Self-control as a moderator of the effects of media use on well-being. In *The Routledge handbook of media use and well-being* (pp. 211-222). Routledge.
- Kleemans, M., Daalmans, S., Carbaat, I., & Anschütz, D. (2018). Picture perfect: The direct effect of manipulated Instagram photos on body image in adolescent girls. *Media Psychology, 21*(1), 93-110. <https://doi.org/10.1080/15213269.2016.1257392>
- Leonhardt, A., Könen, T., Dirk, J., & Schmiedek, F. (2016). How differentiated do children experience affect? An investigation of the within- and between-person structure of children's affect. *Psychological Assessment, 28*(5), 575–585. <https://doi.org/10.1037/pas0000195>
- Li, C., Dang, J., Zhang, X., Zhang, Q., & Guo, J. (2014). Internet addiction among Chinese adolescents: The effect of parental behavior and self-control. *Computers in Human Behavior, 41*, 1–7. <https://doi.org/10.1016/j.chb.2014.09.001>
- Limberger, M. F., Schmiedek, F., Santangelo, P., Reichert, M., Meyer-Lindenberg, A., Tost, H., & Ebner-Priemer, U. W. (under review). Assessing affect in adolescents with e-diaries: Model fit and reliability in multilevel confirmatory factor analyses of different factor models.
- Liu, M., Wu, L., & Yao, S. (2016). Dose–response association of screen time-based sedentary behaviour in children and adolescents and depression: A meta-analysis of observational studies. *British Journal of Sports Medicine, 50*(20), 1252-1258. <http://dx.doi.org/10.1136/bjsports-2015-095084>
- Midgley, C., Thai, S., Lockwood, P., Kovacheff, C. & Page-Gould, E. (2021). When every day is a high school reunion: Social media comparisons and self-esteem. *Journal of Personality and Social Psychology, 121*(2), 285-307. <http://dx.doi.org/10.1037/pspi0000336>
- Molenaar, P. C. M. (2004). A manifesto on psychology as idiographic science: Bringing the person back into scientific psychology, this time forever. *Measurement: Interdisciplinary Research & Perspective, 2*(4), 201–218. https://doi.org/10.1207/s15366359mea0204_1
- Nesi, J., & Prinstein, M. J. (2015). Using social media for social comparison and feedback-seeking: Gender and popularity moderate associations with depressive symptoms. *Journal of Abnormal Child Psychology, 43*(8), 1427–1438. <https://doi.org/10.1007/s10802-015-0020-0>

- Reinecke, L., Gilbert, A., & Eden, A. (2022). Self-regulation as a key boundary condition in the relationship between social media use and well-being. *Current Opinion in Psychology*, *45*, 101296. <https://doi.org/10.1016/j.copsyc.2021.12.008>
- Reis, H. T. (2012). Why researchers should think “Real-world”: A conceptual rationale. In M. R. Mehl & T. S. Conner (Eds.), *Handbook of research methods for studying daily life* (pp. 3–21). New York, NY: Guilford Press.
- Schmidt, A., Dirk, J., & Schmiedek, F. (2019). The importance of peer relatedness at school for affective well-being in children: Between- and within-person associations. *Social Development*, *28*(4), 873–892. <https://doi.org/10.1111/sode.12379>
- Schmidt, A., Neubauer, A. B., Dirk, J., & Schmiedek, F. (2020). The bright and the dark side of peer relationships: Differential effects of relatedness satisfaction and frustration at school on affective well-being in children's daily lives. *Developmental Psychology*, *56*(8), 1532–1546. <https://doi.org/10.1037/dev0000997>
- Schmuck, D., Karsay, K., Matthes, J., & Stevic, A. (2019). “Looking up and feeling down”. The influence of mobile social networking site use on upward social comparison, self-esteem, and well-being of adult smartphone users. *Telematics and Informatics*, *42*, 101240. <https://doi.org/10.1016/j.tele.2019.101240>
- Schwarz, N. (2012). Why researchers should think “Real-time”: A cognitive rationale. In M. R. Mehl & T. S. Conner (Eds.), *Handbook of research methods for studying daily life* (pp. 22–42). New York, NY: Guilford Press.
- Shiffman, S., Stone, A. A., & Hufford, M. R. (2008). Ecological momentary assessment. *Annual Review of Clinical Psychology*, *4*, 1–32. <https://doi.org/10.1146/annurev.clinpsy.3.022806.091415>
- Trull, T. J., & Ebner-Priemer, U. (2014). The role of ambulatory assessment in psychological science. *Current Directions in Psychological Science*, *23*, 466–470. <https://doi.org/10.1177/0963721414550706>
- Valkenburg, P. M., & Peter, J. (2013). The differential susceptibility to media effects model. *Journal of Communication*, *63*(2), 221–243. <https://doi.org/10.1111/jcom.12024>
- Vogel, E. A., Rose, J. P., Roberts, L. R., & Eckles, K. (2014). Social comparison, social media, and self-esteem. *Psychology of Popular Media Culture*, *3*(4), 206–222. <https://doi.org/10.1037/ppm0000047>
- Yang, C. C. (2016). Instagram use, loneliness, and social comparison orientation: Interact and browse on social media, but don't compare. *Cyberpsychology, Behavior, and Social Networking*, *19*(12), 703–708. <http://dx.doi.org/10.1089/cyber.2016.0201>.

30th May 23

Dear Dr Irmer,

Thank you for your patience during the peer-review process. Your manuscript titled "The Daily Reminder That Others are Better off: The Role of Upward Comparisons in the Link Between Children's Everyday Social Media Use and Well-Being" has now been seen by 3 reviewers, and I include their comments at the end of this message. Overall, they are appreciative of the work that has gone into the revision, but still express a few reservations that need to be addressed before we make a final decision on publication.

We therefore invite you to undertake a last minor revise and resubmit your manuscript, along with a point-by-point response to the reviewers. Please highlight all changes in the manuscript text file.

Editorially, we consider it critical you address the issues pertaining to the social comparison scale. After receiving the reviews, we sought additional feedback from Reviewer 2 regarding the scale, and they agree that stronger stronger upward social comparisons is not the best terminology, and that the issues needs to be addressed in revision through precise description, appropriate interpretation, and transparent discussion of limitations under the subheading "Limitations" in the discussion section. We therefore urge you to follow these recommendations.

To facilitate the next steps, we would also ask you to ensure that the manuscript complies with the journal's formatting standards which you can find in this template: <https://www.nature.com/documents/commspsychol-style-formatting-guide-accept.pdf> and checklist: <https://www.nature.com/documents/commspsychol-style-formatting-checklist-article-rr.pdf>.

Please use the following link to submit your revised manuscript, point-by-point response to the referees' comments (which should be in a separate document to any cover letter) and the completed checklist:
[link redacted]

Please do not hesitate to contact me if you have any questions or would like to discuss these revisions further. We look forward to seeing the revised manuscript and thank you for the opportunity to review your work.

Best regards,

Jennifer Bellingtier

Jennifer Bellingtier, PhD
Senior Editor
Communications Psychology

EDITORIAL POLICIES AND FORMATTING

Editorial Policy: [Policy requirements](https://www.nature.com/documents/nr-editorial-policy-checklist.pdf) (Download the link to your computer as a PDF.)

Furthermore, please align your manuscript with our format requirements, which are summarized on the following checklist:

[Communications Psychology formatting checklist](https://www.nature.com/documents/commspsychol-style-formatting-checklist-article-rr.pdf)

and also in our style and formatting guide [Communications Psychology formatting guide](https://www.nature.com/documents/commspsychol-style-formatting-guide-accept.pdf) .

* **CODE AVAILABILITY:** All Communications Psychology manuscripts must include a section titled "Code Availability" at the end of the methods section. In the event of publication, we require that the custom analysis code supporting your conclusions is made available in a publicly accessible repository; at publication, we ask you to choose a repository that provides a DOI for the code; the link to the repository and the DOI will need to be included in the Code Availability statement. Publication as Supplementary Information will not suffice. We ask you to prepare code at this stage, to avoid delays later on in the process.

*** DATA AVAILABILITY:**

All Communications Psychology manuscripts must include a section titled "Data Availability" at the end of the Methods section or main text (if no Methods). More information on this policy, is available at <http://www.nature.com/authors/policies/data/data-availability-statements-data-citations.pdf>.

At a minimum the Data availability statement must explain how the data can be obtained and whether there are any restrictions on data sharing. Communications Psychology strongly endorses open sharing of data. If you do make your data openly available, please include in the statement:

We recommend submitting the data to discipline-specific, community-recognized repositories, where possible and a list of recommended repositories is provided at <http://www.nature.com/sdata/policies/repositories>.

If a community resource is unavailable, data can be submitted to generalist repositories such as [figshare](https://figshare.com/) or [Dryad Digital Repository](http://datadryad.org/). Please provide a unique identifier for the data (for example a DOI or a permanent URL) in the data availability statement, if possible. If the repository does not provide identifiers, we encourage authors to supply the search terms that will return the data. For data that have been obtained from publicly available sources, please provide a URL and the specific data product name in the data availability statement. Data with a DOI should be further cited in the methods reference section.

REVIEWERS' EXPERTISE:

Reviewer #1 Social comparison, experience sampling

Reviewer #2 Development, media use

Reviewer #3 Social comparison, media use

REVIEWERS' COMMENTS:

Reviewer #1 (Remarks to the Author):

Overall, I am satisfied with the revisions that the authors have made. The authors have done a good job of addressing my primary concerns in the original manuscript. However, I still have some smaller

questions and concerns when I read the revised manuscript.

1. On page 6 of the revised manuscript, the authors mentioned person-specific effects, but it is not clear what they mean by person-specific effects. Until this point, the authors have discussed between-person and within-person effects. It would be helpful for the reader to understand the literature, especially one with so many mixed findings if the authors explained what is meant by person-specific effects.

2. On page 7, the authors suggest that Midgley et al. (2021) showed that comparisons occur automatically on physical appearance, popularity, or leisure activities; however, this is inaccurate. Midgley et al. (2021) demonstrate that these are the most common social media comparisons domains, but do not provide any evidence to suggest that these comparisons occur automatically. If the authors would like to discuss the automaticity of social comparisons, I would suggest citing Gilbert et al. (1995).

Gilbert, D. T., Giesler, R. B., & Morris, K. A. (1995). When comparisons arise. *Journal of Personality & Social Psychology*, 69(2), 227–236.

3. On page 9, the authors point out a gap in the literature suggesting that most participants in past studies were above the age of 17 and it is unclear what the results would look like in younger samples. Here, it would be useful to indicate that the authors will discuss the importance of looking at these effects in a younger population later on or to discuss their rationale for examining the younger sample here given that they already point out the gap at this point. It would be natural to then elaborate on the importance of filling this gap.

4. On page 10, the authors mention existing research demonstrating how self-control moderates the effects of media use on well-being. This statement is vague because it is not clear what the effect is. Understanding what past research has specifically found regarding self-control as a moderator of the association between media use on well-being would help readers to better understand why this moderator was selected.

5. On page 16, the authors added that participants were instructed to consider only smartphone use of various social media platforms. Is there a specific reason the authors included this restriction? Is this to make the study comparable to past studies or do the authors believe that scrolling social media on smartphones is different from scrolling social media on other devices such as laptops or computers?

6. Although I still think that the upward social comparison measure the authors have used is flawed in various ways, they should use the terminology used by other social comparison researchers as opposed to creating their own terminology. It is not clear what a “stronger” upward comparison is. Does that mean the comparison was more intense? More vivid? The authors go on to state that by “stronger” they mean that youth perceive a greater discrepancy between themselves and others. In past research describing these discrepancies between the self and comparison others, researchers have described the comparisons as more extreme (see Diel, Grelle, & Hofmann, 2021 and Midgley et al., 2021). Thus, I would suggest that the authors change their terminology to match past research.

7. On page 40, the authors state that their findings are not consistent with Boer et al. (2022). The authors propose that one potential reason for this difference is that Boer et al. (2022) used an older

sample; however, Midgley et al. found this pattern of effects in even an even older sample. Why would the findings disappear and then re-appear for a few years? This explanation for the difference in findings appears to be flawed.

8. On page 41, the authors suggest that interventions aimed at ameliorating the negative effects of social media use on subjective well-being might be effective at targeting upward social comparisons; however, the authors did not draw from the rich literature on social comparisons to come up with any potential suggestions for how to attenuate the negative effects of upward comparisons. One potential intervention that could be suggested is examining the effect of self-affirmation. Past research (Spencer et al., 2001) has shown that self-affirmation helps individuals cope with threats to the self. Thus, one potential intervention could be a self-affirmation intervention.

Spencer, S. J., Fein, S., & Lomore, C. D. (2001). Maintaining one's self-image vis-a'-vis others: The role of self-affirmation in the social evaluation of the self. *Motivation and Emotion*, 25, 41-65.

Reviewer #2 (Remarks to the Author):

Thank you again for the opportunity to read this revised paper. I would like to thank the authors for their attention to my comments and those of the other reviewers. After reading the revised draft, I have a few additional comments for the authors, but note that they are minor in nature. I present them in the order in which they appear in the manuscript.

1. Page 4, three lines up from the bottom: there should be no comma after the word 'both'.
2. In reading the section claiming few research studies on social media use and well-being among adolescents that employ within- and between-subjects analyses, I realized the authors are omitting a recent study by Scheurs and Vandenbosch, published in *Telematics and Informatics*. I think it would be helpful to weave this in.
<https://www.sciencedirect.com/science/article/abs/pii/S0736585322000983>
3. Page 6: following the citations of Beyens et al. (2020; 2021), I think it would be helpful to include a couple of lines describing these findings in more detail and connecting them to the present study.
4. Page 7, middle of the page: following 'pleasant information', please change 'are' to 'is'.
5. First line of page 8: what does it mean to be 'extremely' upward?
6. In general, in this revision I noticed many long paragraphs (more than one page of text). Please revise for enhanced readability.
7. Page 10: the authors write that high-intense users experience stronger associations between social media use and well-being – do they mean ill-being?
8. page 12, midway down the page: I think it would be better to change 'era' with 'developmental period'.
9. page 13: The authors write that there is evidence that the associations examined in this work extend to multi-platform data --- what does this mean? I was not clear on this.
10. For the affective well-being measure, the authors write that there are nine items, but then only list seven.
11. Were participants only reminded to consider their smartphone based social media use in the pre-test, or throughout? I would think that adolescents might have a hard time remembering this if it was only mentioned once, and if so, this could probably be a limitation.
12. page 18, line 2: please change individual's to individuals'
13. In the results, I realized that the authors considered the subscales of the social comparison

measure separately, rather than together. Is there a theoretical reason for this? If so, please add that in for justification.

14. page 39: please change 'insignificant' to 'non-significant'.

Reviewer #3 (Remarks to the Author):

Thank you for responding to my comments.

I still don't think the scale was really measuring upward social comparison. An item of upward social comparison would be "Today, I compared myself with those who seemed to have a better life than me." This would be more consistent with the definition the authors provided, citing Gerber et al. (2018). Another reviewer appears to share a similar concern in the scale used. I would like the authors to at least acknowledge this as a limitation in the manuscript.

Communications Psychology

June 30, 2023

Dear Reviewers,

Thank you very much for your feedback to the revised version of our manuscript and the helpful comments.

For you to be able to follow the changes we made, we highlighted edit parts of the manuscript using blue font. In our responses, we inserted the new or adjusted paragraphs and included the page number to indicate precisely where we implemented changes. Please note that in doing the revision, we formatted our manuscript according to the guidelines provided by the Journal. Hence, we adapted several sections and added further information (e.g., changed the title, changed the Heading for 2.4, changed the citation style, and added a statement that the study was not preregistered).

In the following, we describe point by point how we responded to your specific concerns:

Responses to Reviewer 1's comments:

R1.0 Overall, I am satisfied with the revisions that the authors have made. The authors have done a good job of addressing my primary concerns in the original manuscript. However, I still have some smaller questions and concerns when I read the revised manuscript.

RESPONSE: Thank you for the positive feedback on the revisions we made.

R1.1 On page 6 of the revised manuscript, the authors mentioned person-specific effects, but it is not clear what they mean by person-specific effects. Until this point, the authors have discussed between-person and within-person effects. It would be helpful for the reader to understand the literature, especially one with so many mixed findings if the authors explained what is meant by person-specific effects.

RESPONSE: Effects on a within-person level refer to associations that unfold within individuals across time. Hence, we receive one estimate for the within-person association of variables for the whole sample, implying that the estimate refers to an aggregate of the single within-person effect sizes of all persons. By contrast, person-specific effects refer to within-person effect sizes that are calculated for each individual separately. We added further information on this on pp. 5-6:

“Likewise, Beyens et al.³³ and Beyens et al.³⁸ did not find Instagram or social media use to be associated with adolescents’ affective well-being on a within-person level. Aiming to explore the relations in more detail, the authors examined person-specific effects, that is, they calculated individual within-person effect sizes separately for each adolescent. These analyses revealed differences between adolescents in the significance and direction of the associations between social media use and affective well-being. Thereby, most adolescents showed non-significant relations, while some adolescents showing increased or decreased well-being when using social media³³. After having found a non-significant within-person effect of social media use on self-esteem, Valkenburg et al.³⁹ also investigated person-specific effects. They showed that the majority of adolescents (88%) experienced no (or very small) effects of social media use on self-esteem, while only few adolescents experienced positive (4%) or negative (8%) effects.”

R1.2 On page 7, the authors suggest that Midgley et al. (2021) showed that comparisons occur automatically on physical appearance, popularity, or leisure activities; however, this is

inaccurate. Midgley et al. (2021) demonstrate that these are the most common social media comparisons domains, but do not provide any evidence to suggest that these comparisons occur automatically. If the authors would like to discuss the automaticity of social comparisons, I would suggest citing Gilbert et al. (1995).

Gilbert, D. T., Giesler, R. B., & Morris, K. A. (1995). When comparisons arise. *Journal of Personality & Social Psychology*, 69(2), 227–236.

RESPONSE: We removed the word “automatically” from the sentence referred to by the Reviewer.

R1.3 On page 9, the authors point out a gap in the literature suggesting that most participants in past studies were above the age of 17 and it is unclear what the results would look like in younger samples. Here, it would be useful to indicate that the authors will discuss the importance of looking at these effects in a younger population later on or to discuss their rationale for examining the younger sample here given that they already point out the gap at this point. It would be natural to then elaborate on the importance of filling this gap.

RESPONSE: We agree with the Reviewer and decided to move the later discussion of the importance of examining younger populations (part of the section “The Present Study”) to the mentioned paragraph on page 9.

R1.4 On page 10, the authors mention existing research demonstrating how self-control moderates the effects of media use on well-being. This statement is vague because it is not clear what the effect is. Understanding what past research has specifically found regarding self-control as a moderator of the association between media use on well-being would help readers to better understand why this moderator was selected.

RESPONSE: We thank the Reviewer for pointing out that more information is needed on this topic. We have revised the section on why we investigated the failure of self-control as a potential moderator by explaining findings of previous research in more detail and deriving our expectations (pp. 10-11). In response to this comment, we also explicated our expectations for the other moderators (p. 11).

“Besides gender, prior studies with adults also motivated us to examine self-control failure with regard to social media use. Self-control refers to the “ability to override or change one’s inner responses, as well as to interrupt undesired behavioral tendencies (such as impulses) and refrain from acting on them”^{65(p.274)}. Hence, self-control failure related to social media use assesses the degree to which individuals use social media platforms although it stands in conflict with other goals or tasks, or with using time efficiently⁶⁶. Social media-related self-control failure is associated with deficient self-regulation⁶⁶ and can be predictive of later social media addiction⁶⁷. Previous work that has reviewed existing evidence with individuals of all ages concludes that self-control and the failure thereof serve as significant moderators of the effects of media use on well-being^{68,69}. Thus, the effect of social media use on well-being depends on individual’s degree of self-control failure: Failure to self-control social media use can impair subjective well-being by increasing negative emotions following social media use (e.g., guilt) as well as by decreasing the beneficial impact of social media use, that is, by reducing the experience of positive emotions (e.g., enjoyment, vitality)^{68,70}. Therefore, we decided to explore whether social media self-control failure moderated the associations of interest in the present study, assuming children with higher self-control failure to show stronger links between social media use and negative self-worth and negative affect, and to show weaker links between social media use and positive self-worth and positive affect than children with lower self-control failure.”

R1.5 On page 16, the authors added that participants were instructed to consider only smartphone use of various social media platforms. Is there a specific reason the authors included this restriction? Is this to make the study comparable to past studies or do the authors believe

that scrolling social media on smartphones is different from scrolling social media on other devices such as laptops or computers?

RESPONSE: It is correct that we instructed participants to only consider smartphone social media use. We decided for this procedure, as another aim of the current project was to investigate the congruence of a subjective assessment of social media use (as presented in the present manuscript: “How much did you use Instagram today?”) with an objective assessment (i.e., time spent on each social media platform every day, reported in minutes). Thereby, participants retrieved the objective time assessment from their smartphones, that is, the objective measure referred to smartphone social media use only. To be able to compare the subjective to the objective measure, the subjective measure hence needed to also refer to smartphone use. This research question was motivated from previous research has shown that subjective time estimates of social media use show rather low accuracy as compared to objective time measures and is examined as part of another manuscript that is accepted for publication (but not yet available online). We now refer to this paper on p. 15.

R1.6 Although I still think that the upward social comparison measure the authors have used is flawed in various ways, they should use the terminology used by other social comparison researchers as opposed to creating their own terminology. It is not clear what a “stronger” upward comparison is. Does that mean the comparison was more intense? More vivid? The authors go on to state that by “stronger” they mean that youth perceive a greater discrepancy between themselves and others. In past research describing these discrepancies between the self and comparison others, researchers have described the comparisons as more extreme (see Diel, Grelle, & Hofmann, 2021 and Midgley et al., 2021). Thus, I would suggest that the authors change their terminology to match past research.

RESPONSE: We changed the term “stronger upward social comparisons” into “more extreme upward social comparisons” throughout the manuscript. We also changed related terms such as “...stronger impression that others had a better life ...” into “...more extreme impression that others had a better life...”.

Furthermore, in response to this comment and comment R3.1, we added another limitation on this topic (p. 39):

“Fourth, we developed our items assessing upward social comparisons based on the work by Boer et al.²³. However, it may be argued that the items do not exactly tap the construct of engaging in upward social comparisons, but rather the impression or feeling that results from it. Therefore, it would be interesting to replicate our study with a different measure including items such as “Today, I compared myself to those who seemed to have a better life than me” and “Today, I compared myself to those who seemed to be prettier than me”, and to compare such a measure to the present scale.”

R1.7 On page 40, the authors state that their findings are not consistent with Boer et al. (2022). The authors propose that one potential reason for this difference is that Boer et al. (2022) used an older sample; however, Midgley et al. found this pattern of effects in even an even older sample. Why would the findings disappear and then re-appear for a few years? This explanation for the difference in findings appears to be flawed.

RESPONSE: We revised the paragraph the Reviewer referred to, removed the age-related explanation, and added another possible explanation for diverging results (pp. 35-36):

“Possible reasons for these inconsistencies could be the use of different measures of social media use and subjective well-being or differences in study design. Hence, Boer et al.²³ conducted a longitudinal study and found problematic social media use to predict increases in upward social comparisons over time. However, they did not find these increases in upward social comparisons to predict decreases in mental health one year later, rejecting the expected mediation hypothesis. Based on their findings, the authors suggested that “there may have been a mediating effect, but the measurements were possibly

too far apart to observe it^{23(p.9)}, which is why they called for studies “using more intensive longitudinal data, such as daily measures of SMU [social media use] and mental health”^{23(p.10)}. Following this recommendation, the present work tested a potential mediation based on data being collected every day across two weeks. Thus, differences in the time intervals between measurements may have contributed to diverging results here.”

R1.8 On page 41, the authors suggest that interventions aimed at ameliorating the negative effects of social media use on subjective well-being might be effective at targeting upward social comparisons; however, the authors did not draw from the rich literature on social comparisons to come up with any potential suggestions for how to attenuate the negative effects of upward comparisons. One potential intervention that could be suggested is examining the effect of self-affirmation. Past research (Spencer et al., 2001) has shown that self-affirmation helps individuals cope with threats to the self. Thus, one potential intervention could be a self-affirmation intervention.

Spencer, S. J., Fein, S., & Lomore, C. D. (2001). Maintaining one’s self-image vis-a-vis others: The role of self-affirmation in the social evaluation of the self. *Motivation and Emotion*, 25, 41-65.

RESPONSE: Thank you for suggesting self-affirmations as a concrete example for potential interventions. We are happy to include this in the Discussion on p. 37:

“One potential intervention could be a self-affirmation intervention that builds upon prior work showing that self-affirmations help individuals cope with threats to the self¹⁰¹. According to self-affirmation theory¹⁰², individuals can buffer themselves against threats to the self by thinking about self-resources, that is, positive aspects of their self that may or may not be related to the threatened aspect. As upward social comparisons automatically pose a threat to the self, it may be promising to test whether self-affirmations can mitigate the negative effects of social media use on subjective well-being, specifically by ameliorating the negative impact of upward social comparisons on self-worth.”

Responses to Reviewer 2’s comments:

R2.0 Thank you again for the opportunity to read this revised paper. I would like to thank the authors for their attention to my comments and those of the other reviewers. After reading the revised draft, I have a few additional comments for the authors, but note that they are minor in nature. I present them in the order in which they appear in the manuscript.

RESPONSE: We thank the Reviewer for the positive feedback on our revised paper.

R2.1 Page 4, three lines up from the bottom: there should be no comma after the word ‘both’.

RESPONSE: We thank the Reviewer for noticing this and removed the comma after ‘both’.

R2.2 In reading the section claiming few research studies on social media use and well-being among adolescents that employ within- and between-subjects analyses, I realized the authors are omitting a recent study by Scheurs and Vandenbosch, published in *Telematics and Informatics*. I think it would be helpful to weave this in.

<https://www.sciencedirect.com/science/article/abs/pii/S0736585322000983>

RESPONSE: Thank you for suggesting the interesting work by Schreurs and Vandenbosch. After having read the paper carefully, we decided to not include it in our manuscript. In their paper, the authors examined the link between social media literacy (i.e., knowledge about the positivity-biased nature of social media content), own positivity-based behaviors (i.e., posting idealized and unauthentic content on social media), and self-esteem in adolescents. Data was collected at three time points across

a four-month interval, hence, the study had a larger three-wave panel design. Although within-person associations were examined, the study did not employ an intensive longitudinal design that would include at least one assessment per day across several days (i.e., daily diary design or experience sampling method). In the part of our manuscript that the Reviewer refers to, we cite within-person studies that investigated such relatively short-term fluctuations in social media use and subjective well-being and employed a daily diary or experience sampling design (mostly combined with an ambulatory assessment approach that refers to data collection in individuals' everyday lives and, hence, natural contexts). Furthermore, we focus rather on the general use of social media per se than on positivity-based posting behavior, in particular. Therefore, we decided to not include the study in our manuscript.

R2.3 Page 6: following the citations of Beyens et al. (2020; 2021), I think it would be helpful to include a couple of lines describing these findings in more detail and connecting them to the present study.

RESPONSE: We agree with the Reviewer and added more information as well as a connection to the current study on pp. 5-6. In describing the person-specific effects of Beyens et al. in more detail, we added another relevant study (Valkenburg et al., 2021). We now also refer to this study in the Discussion on p. 32.

“Likewise, Beyens et al.³³ and Beyens et al.³⁸ did not find Instagram or social media use to be associated with adolescents' affective well-being on a within-person level. Aiming to explore the relations in more detail, the authors examined person-specific effects, that is, they calculated individual within-person effect sizes separately for each adolescent. These analyses revealed differences between adolescents in the significance and direction of the associations between social media use and affective well-being. Thereby, most adolescents showed non-significant relations, while some adolescents showing increased or decreased well-being when using social media³³. After having found a non-significant within-person effect of social media use on self-esteem, Valkenburg et al.³⁹ also investigated person-specific effects. They showed that the majority of adolescents (88%) experienced no (or very small) effects of social media use on self-esteem, while only few adolescents experienced positive (4%) or negative (8%) effects. Supporting heterogeneity in the effects, Boer et al.³⁴ also showed that within-person associations between social media use and life satisfaction ranged from negative to positive across adolescents.

Altogether, previous studies demonstrate that individuals differ in their effects of social media use on subjective well-being. Hence, there is a strong need for research investigating why some adolescents seemingly benefit from using social media, while others are harmed by it and yet others seem to be unaffected^{34,38}. As emphasized by Valkenburg et al.⁸, examining mediators and moderators in the associations between social media use and well-being might help to shed light on this heterogeneity. The present study follows up on this by examining upward social comparisons as a mediator of the link between daily social media use and daily subjective well-being in children and young adolescents. Furthermore, potential moderators of the associations of interest are explored.”

“However, our result of significant within-person associations between social media use and self-worth stand in contrast to the results by Valkenburg et al.³⁹, who did not find a significant within-person effect of social media use on self-esteem. Yet, the authors assessed the use of Instagram, WhatsApp, and Snapchat; hence, only one platform overlapped with the present study and the sample consisted of older participants (13 to 15 years) than the current sample (10 to 14 years). Nonetheless, it is not clear why the current results diverge this much from the findings by Valkenburg et al.³⁹”

R2.4 Page 7, middle of the page: following ‘pleasant information’, please change ‘are’ to ‘is’.

RESPONSE: Thank you, we changed ‘are’ to ‘is’ in this sentence.

R2.5 First line of page 8: what does it mean to be ‘extremely’ upward?

RESPONSE: We clarified how extremity of upward social comparisons is meant in the study we referred to (p. 7):

“Furthermore, the authors showed that social comparisons that were made while using social media were more extremely upward (i.e., were to others being “much-better off”) than social comparisons occurring in different contexts.”

R2.6 In general, in this revision I noticed many long paragraphs (more than one page of text). Please revise for enhanced readability.

RESPONSE: We re-read all paragraphs that were changed as part of the first revision. In this course, we shortened several sentences (e.g., p. 10, p. 17) or made two single shorter sentences out of many long sentences (e.g., p. 3, p. 11, p. 17, p. 38).

R2.7 Page 10: the authors write that high-intense users experience stronger associations between social media use and well-being – do they mean ill-being?

RESPONSE: We apologize for not having been clear on this. In doing the revision, we modified this section. We now explicate the results of previous studies and why we investigated self-control failure more clearly (pp. 10-11):

“Besides gender, prior studies with adults also motivated us to examine self-control failure with regard to social media use. Self-control refers to the “ability to override or change one’s inner responses, as well as to interrupt undesired behavioral tendencies (such as impulses) and refrain from acting on them”^{65(p.274)}. Hence, self-control failure related to social media use assesses the degree to which individuals use social media platforms although it stands in conflict with other goals or tasks, or with using time efficiently⁶⁶. Social media-related self-control failure is associated with deficient self-regulation⁶⁶ and can be predictive of later social media addiction⁶⁷. Previous work that has reviewed existing evidence with individuals of all ages concludes that self-control and the failure thereof serve as significant moderators of the effects of media use on well-being^{68,69}. Thus, the effect of social media use on well-being depends on individual’s degree of self-control failure: Failure to self-control social media use can impair subjective well-being by increasing negative emotions following social media use (e.g., guilt) as well as by decreasing the beneficial impact of social media use, that is, by reducing the experience of positive emotions (e.g., enjoyment, vitality)^{68,70}. Therefore, we decided to explore whether social media self-control failure moderated the associations of interest in the present study, assuming children with higher self-control failure to show stronger links between social media use and negative self-worth and negative affect, and to show weaker links between social media use and positive self-worth and positive affect than children with lower self-control failure.”

R2.8 page 12, midway down the page: I think it would be better to change ‘era’ with ‘developmental period’.

RESPONSE: We agree with the Reviewer and replayed ‘era’ by ‘developmental period’.

R2.9 page 13: The authors write that there is evidence that the associations examined in this work extend to multi-platform data --- what does this mean? I was not clear on this.

RESPONSE: We are sorry for not having explained this properly. We extended the sentence as follows (p. 12):

“However, as there is evidence that the associations examined in this work extend to multi-platform data (i.e., considering the use of different platforms simultaneously), it has also been argued that moving away from single-platform data (i.e., Instagram use only) is essential in order to generalize findings⁷⁶”

R2.10 For the affective well-being measure, the authors write that there are nine items, but then only list seven.

RESPONSE: Thank you very much for bringing this error to our attention. It is correct that we assessed affective well-being with the seven items that we listed in this paragraph. We replaced 'nine' by 'seven'.

R2.11 Were participants only reminded to consider their smartphone based social media use in the pre-test, or throughout? I would think that adolescents might have a hard time remembering this if it was only mentioned once, and if so, this could probably be a limitation.

RESPONSE: The whole study has been framed as being about smartphone use from the beginning. Hence, in addition to subjective use of social media, we also assessed objective social media use by instructing participants to navigate to the screen time application implemented in their smartphone (or to an application installed for the purpose of the study) and to type in the number of minutes they had used each of the social media platforms of interest on that day. This instruction played a major role in the instruction video and participants had the opportunity to re-watch these instructions every day as part of the daily questionnaire. This way, participants were reminded to only consider smartphone social media use in every questionnaire. The objective measure as well as an examination of the congruence of the subjective and objective measures are part of another manuscript that was recently accepted (but is not yet available online). We now refer to this study on p. 15.

R2.12 page 18, line 2: please change individual's to individuals'

RESPONSE: We thank the Reviewer for noticing this error and replaced "individual's" by "individuals".

Responses to Reviewer 3's comments:

R3.0 Thank you for responding to my comments.

RESPONSE: We thank the Reviewer for their helpful comments to our work.

R3.1 I still don't think the scale was really measuring upward social comparison. An item of upward social comparison would be "Today, I compared myself with those who seemed to have a better life than me." This would be more consistent with the definition the authors provided, citing Gerber et al. (2018). Another reviewer appears to share a similar concern in the scale used. I would like the authors to at least acknowledge this as a limitaiton in the manuscript.

RESPONSE: We appreciate the constructive feedback of the Reviewer. We completely agree that the item the Reviewer mentioned assesses upward social comparisons. We developed our items mainly based on the study by Boer et al. (2021, *Computers in Human Behavior*). In their article, the authors wrote: "Respondents indicated, when viewing their peers' messages, photos, or movies on social network sites, how often they thought 'He or she does more fun things than I do', 'He or she has more friends than I do', 'He or she is more popular dan me', 'He or she received more 'likes' than me', and 'He or she looks better than I do' (1 never to 5 very often)." (p. 4). Our items are quite similar to these items, even though we did not refer to specific persons as did Boer et al. (2021) ('He or she ...'):

Today, I had the feeling that others have a better life than me.

Today, I had the feeling that others are happier than I am.

Today, I had the feeling that others are more popular than I am.

Today, I had the feeling that others are prettier than me.

Today, I had the feeling that others were doing more or cooler things than me.

Today, I had the feeling that others have more or cooler stuff than me.

We agree that we assessed the general impression ('feeling') of others being superior in some way and not the absence or presence of engaging in upward social comparisons. However, individuals would not agree to our items if they had not been engaging in upward social comparisons and, hence, have the feeling that others are better off. Nonetheless, we added this as a limitation on p. 39:

“Fourth, we developed our items assessing upward social comparisons based on the work by Boer et al.²³. However, it may be argued that the items do not exactly tap the construct of engaging in upward social comparisons, but rather the impression or feeling that results from it. Therefore, it would be interesting to replicate our study with a different measure including items such as “Today, I compared myself to those who seemed to have a better life than me” and “Today, I compared myself to those who seemed to be prettier than me”, and to compare such a measure to the present scale.”

Boer, M., Stevens, G. W., Finkenauer, C., Looze, M. E. de, & Van den Eijnden, R. J. (2021). Social media use intensity, social media use problems, and mental health among adolescents: Investigating directionality and mediating processes. *Computers in Human Behavior, 116*, 106645. <https://doi.org/10.1016/j.chb.2020.106645>

4th Jul 23

Dear Dr Irmer,

Your manuscript titled "The Mediating Role of Upward Comparisons in the Link Between Children's Daily Social Media Use and Well-Being" and response to the reviewers has now been editorially evaluated. I am delighted to say that we are happy, in principle, to publish a suitably revised version in *Communications Psychology* under the open access CC BY license (Creative Commons Attribution v4.0 International License).

We therefore invite you to revise your paper one last time to address our editorial requests. At the same time we ask that you edit your manuscript to comply with our format requirements and to maximise the accessibility and therefore the impact of your work.

EDITORIAL REQUESTS:

**Please fully report all statistics, as detailed in the attached checklist; for example, it is not sufficient to report just Pseudo R without any further details.

**All statistics incorporated in the Figures need to be reported redundantly in the main text.

**Statistics also need to be reported in full in the Tables, i.e., include the degrees of freedom, test statistic, effect size measure (if different from the test-statistics), confidence interval, and exact p-value.

** Currently, you repeat many arguments and reflections between Introduction and Discussion. Please streamline the text to avoid unnecessary repetition across sections. For example, you do not need to start the Discussion with a repetition of study rationale and previous findings. Instead, you might want to briefly highlight your most relevant results.

**Relatedly, the Discussion itself contains many unnecessary repetitions. It is not necessary to repeatedly point out the same features of the study that makes it distinct from previous work across (for example, repeatedly pointing out the sample was younger/ the internet platforms differed). Streamlining the text would increase readability without information getting lost.

SUBMISSION INFORMATION:

In order to accept your paper, we require the files listed at the end of the Editorial Requests Table; the list of required files is also available at <https://www.nature.com/documents/commsj-file->

checklist.pdf .

OPEN ACCESS:

Communications Psychology is a fully open access journal. Articles are made freely accessible on publication under a [CC BY license](http://creativecommons.org/licenses/by/4.0) (Creative Commons Attribution 4.0 International License). This license allows maximum dissemination and re-use of open access materials and is preferred by many research funding bodies.

For further information about article processing charges, open access funding, and advice and support from Nature Research, please visit <https://www.nature.com/commspsychol/article-processing-charges>

At acceptance, you will be provided with instructions for completing this CC BY license on behalf of all authors. This grants us the necessary permissions to publish your paper. Additionally, you will be asked to declare that all required third party permissions have been obtained, and to provide billing information in order to pay the article-processing charge (APC).

*** TRANSPARENT PEER REVIEW:** Communications Psychology uses a transparent peer review system. On author request, confidential information and data can be removed from the published reviewer reports and rebuttal letters prior to publication. If you are concerned about the release of confidential data, please let us know specifically what information you would like to have removed. Please note that we cannot incorporate redactions for any other reasons.

*** CODE AVAILABILITY:** All Communications Psychology manuscripts must include a section titled "Code Availability" at the end of the methods section. We require that the custom analysis code supporting your conclusions is made available in a publicly accessible repository at this stage; please choose a repository that generates a digital object identifier (DOI) for the code; the link to the repository and the DOI must be included in the Code Availability statement. Publication as Supplementary Information will not suffice.

*** DATA AVAILABILITY:**

[link redacted]

****** This url links to your confidential home page and associated information about manuscripts you may have submitted or be reviewing for us. If you wish to forward this email to co-authors, please

delete the link to your homepage first **

Best regards,

Jennifer Bellingtier

Jennifer Bellingtier, PhD
Senior Editor
Communications Psychology